# Anchor Data Augmentation

**Nora Schneider**[1]   **Shirin Goshtasbpour**[1,2]   **Fernando Perez-Cruz** [1,2]

[1]Computer Science Department, ETH Zurich, Zurich, Switzerland
[2]Swiss Data Science Center, Zurich, Switzerland

nschneide@student.ethz.ch
shirin.goshtasbpour@inf.ethz.ch
fernando.perezcruz@sdsc.ethz.ch

## Abstract

We propose a novel algorithm for data augmentation in nonlinear over-parametrized regression. Our data augmentation algorithm borrows from the literature on causality and extends the recently proposed Anchor regression (AR) method for data augmentation, which is in contrast to the current state-of-the-art domain-agnostic solutions that rely on the Mixup literature. Our Anchor Data Augmentation (ADA) uses several replicas of the modified samples in AR to provide more training examples, leading to more robust regression predictions. We apply ADA to linear and nonlinear regression problems using neural networks. ADA is competitive with state-of-the-art C-Mixup solutions. [1]

## 1  Introduction

Data augmentation is one of the key ingredients of any successful application of a machine learning classifier. The first example that typically comes to mind is the in-depth description of the data augmentation in the now-famous Alexnet paper [26]. Data augmentation algorithms come in different flavors, and they mostly rely on the expectation that small perturbations, invariances, or symmetries applied to the input will not change the class label. That way, we can present 'fresh new' samples as alterations of the available examples for training. These transformations modify the input distribution to make the algorithm more robust for cases where the distribution of the test set may differ from that of the training set. We refer the reader to the related work section (Section 2.1) for an overview and description of different data augmentation strategies.

The literature for data augmentation in regression is slim. The paper on Mixup augmentation [51] proposes a simple and general scheme for data augmentation using convex combinations of samples. The authors only apply their data augmentation proposal to classification problems. They conjecture in the discussion that the application to regression is *straightforward*, however, this is not the case in practice. Mixup is theoretically analyzed in [5, 52] as a regularization technique for classification and regression problems. However, it is only illustrated in classification problems.

The Mixup algorithm has been extended to regression problems in [18, 49], in which the authors explain that Mixup cannot be blindly applied to regression problems. To our knowledge, these are the only two papers in which data augmentation for regression is proposed. RegMix [18] relies on a hard-to-train prior neural network controller before augmenting the data using a Mixup strategy. *C-Mixup* [49], a method proposed more recently, solves some of the issues limiting the standard Mixup algorithm for regression problems. The authors propose to mix only closeby samples in the

---

[1]Our Python implementation of ADA is available at: https://github.com/noraschneider/anchordataaugmentation/

37th Conference on Neural Information Processing Systems (NeurIPS 2023).

output space (i.e., samples which have close enough labels). This strategy is only valid when the target variables are monotonic with the input and is applied in a transformed space. The authors present comprehensive results in data augmentation for in-distribution generalization, task generalization and out-of-distribution robustness.

In this paper, we rely on the causality literature to provide a different avenue for augmenting data in regression problems. Causal discovery finds the causes of a response variable among a given set of observations or helps to recognize the causal relations between a set of variables [39]. These causes allow us to understand how these relations will change if we were to intervene in a subset of the (input) variables or what would be the effect on the output. So, in general, the regression model will be robust to perturbations in the input variables making the prediction less sensitive to changes in the distribution of the test set. For example, the authors in [40] use the invariance property for prediction to perform causal inference. In turn, Anchor Regression (AR) builds upon the causality literature to obtain robust regression solutions when the input variables have been perturbed [42]. The procedure relies on anchor variables capturing the heterogeneity within a dataset and a parameter $\gamma$ that measures the deviation with respect to the least square solution. Once the values of the anchors are known, AR modifies the data and obtains the least square solution, as detailed in Section 2.2.

In this paper, we propose **Anchor Data Augmentation (ADA)** to augment the training dataset with several replicas of the available data. We use a simple clustering of the data to encode a homogeneous group of observations and use different values of $\gamma$ to robustify the solution to different strengths of potential distribution shifts. In every minibatch, we sample $\gamma$ from a predetermined range around $\gamma = 1$. As AR was developed for linear regression, the data augmentation strategy needs to be modified for nonlinear regression accordingly. We validate ADA for in-distribution generalization and out-of-distribution robustness under the same conditions proposed in C-Mixup [49], as well as some illustrative linear and nonlinear regression examples. In the replicated experiments, ADA is competitive or superior to other augmentation strategies such as C-Mixup, although on some datasets the performance gain is marginal.

The rest of the paper is organized as follows: First, we provide background information in Section 2. We give a brief overview of related work on data augmentation in Section 2.1 and summarize the key concepts on Anchor Regression in Section 2.2. Second, Section 3 shows how we extend Anchor Regression and introduces ADA. Section 4 reports empirical evidence that our approach can improve predictions, especially in over-parameterized settings. We conclude the paper in Section 5.

## 2 Background

### 2.1 Data Augmentation

Many different data augmentation methods have been proposed in recent years with several applications in mind. Still most augmentations we mention here use human-designed transformations based on domain knowledge which leave the target variable invariant. For instance, Cutout [10] is an image-specific augmentation technique that is successfully used to train models on CIFAR10 and CIFAR100 [25], but was determined to be unsuitable for larger image datasets like ImageNet with higher resolution [9]. Other augmentation methods for images such as random crop, horizontal or vertical mirroring, random rotation, or translation [29, 43] may similarly apply to a certain group of image datasets while being inapplicable to others, e.g. datasets of digits and letters.

In an attempt to automate the augmentation process and reduce human involvement, policy or search-based automated augmentation methods were developed. In AutoAugment [7] a neural network is trained with Reinforcement Learning (RL) to combine an assortment of transformations in varying strengths to apply on samples of a given dataset and improve the model accuracy. Methods such as RandAugment [8], Fast AutoAugment [30], UniformAugment [32] and TrivialAugment [36] aim at reducing the cost of the pretraining search phase in automated augmentation with randomized transformations and reduced search space.

Alternatively, in order to adapt the augmentation policy to the model during training, Population-Based Augmentation [16] and Online Hyperparameter Learning [31] use multiple data augmentation workers that are updated using evolutionary strategies and RL, respectively. Adversarial AutoAugment [53] and AugMax [47] optimize for the augmentation policy that deteriorates the training accuracy

and improves its robustness. DivAug [34] finds the policy which maximizes the diversity of the augmented data.

Having a separate search phase for optimal augmentation policy is computationally expensive and may exceed the required computation to train the downstream model [8, 48]. In addition, these methods and their online counterparts need to be trained separately on every single dataset. While OnlineAugment [44] and DDAS exploit meta-learning to avoid this problem, they still rely on a set of predefined class invariant transformations that require domain-specific information.

Generic transformations such as Gaussian or adversarial noise [10, 28, 45] and dropout [3] are also effective in expanding the training dataset. Generative models such as Generative Adversarial Networks (GAN) [13] and Variational Auto-Encoders (VAE) [22] are trained in [1, 6, 44] to synthesize samples close to the low dimensional manifold of the data for classification.

Mixup [51] is a popular data augmentation using a convex combination of pairs of samples from different classes and their softened labels for augmentation. Mixup is only evaluated on classification problems, even though it is claimed that the application to regression is straightforward. Various extensions of Mixup have been proposed to prevent data manifold intrusion [46], use more complex mixing strategies [33, 50] or account for saliency in augmented samples [20, 21]. These methods were predominantly designed to excel in classification tasks. In particular, Mixup for regression was studied in [5, 18, 49, 52] but it was reported to adversely impact the predictions in regression problems when misleading augmented samples are generated from a pair of faraway samples.

## 2.2 Anchor Regression

We summarize the key concepts of Anchor Regression (AR) as presented in [42]. Let $X \in \mathcal{X}$ and $y \in \mathcal{Y}$ be the predictors and target variables sampled from distribution $(X, y) \sim P_{\text{train}}$, $\mathcal{X} \subseteq \mathbb{R}^d$ and $\mathcal{Y} \subseteq \mathbb{R}$. Traditionally, a causal framework models the relation of $y$ and $X$ to accurately predict the value of $y$ under given interventions or arbitrary perturbations on $X$. A commonly held assumption is that the underlying causal relation among variables remains the same while the sampling distribution $P_{\text{train}}$ is altered by the intervention shift or the applied perturbation. For instance, if the distribution $P_{\text{train}}$ is induced by an unknown linear causal model, then the causally optimal parameters can be expressed as the solution to the optimization problem:

$$b_{\text{causal}} = \arg\min_b \max_{P \in \mathcal{P}} \mathbb{E}_P[(y - X^T b)^2], \tag{1}$$

where $\mathcal{P}$ is the class of distributions containing all interventions on components of $X$ [41]. Therefore, causal parameters provide distributionally robust predictions that are optimal under the intervention in $\mathcal{P}$. In comparison, Ordinary Least Squares (OLS):

$$b_{\text{OLS}} = \arg\min_b \mathbb{E}_{P_{\text{train}}}[(y - X^T b)^2], \tag{2}$$

may lead to arbitrarily large predictive errors on distributions in $\mathcal{P}$. On the other hand, on $P_{\text{train}}$, causal parameters $b_{\text{causal}}$ lead to conservative predictions, while $b_{\text{OLS}}$ presents optimal least squared performance.

To trade-off predictive accuracy on the training distribution with distribution robustness and to enforce stability over statistical parameters, AR [4, 42] proposes to relax the regularization in the optimization problem in (1) to a smaller class of distributions $\mathcal{P}$.

Assume that $X$ and $y$ are centered and have finite variance. We use $A \in \mathbb{R}^q$ (called anchors) to denote the exogenous (random) variables in $X$ which generate heterogeneity in $y$. We further denote the $L_2$-projection on the linear span of the components of $A$ with $P_A$ and $\text{Id}(y) = y$. Under linear assumption between $A$ and $(X, y)$, we can write the relaxed optimization problem as:

$$b_{\gamma, A} = \arg\min_b \mathbb{E}_{P_{\text{train}}}[((\text{Id} - P_A)(y - X^T b))^2] + \gamma \mathbb{E}_{P_{\text{train}}}[(P_A(y - X^T b))^2], \tag{3}$$

where $\gamma > 0$ is a hyperparameter. The first term of the AR objective in Equation 3 is the loss after "partialling out" the anchor variable, which refers to first linearly regressing out $A$ from $X$ and $y$ and subsequently using OLS on the residuals. The second term is the well-known estimation objective used in the Instrumental Variable setting [11]. Therefore, for different values of $\gamma$ AR interpolates between the partialling out objective ($\gamma = 0$) and the IV estimator ($\gamma \to \infty$) and coincides with

OLS for $\gamma = 1$. The authors show that the solution of AR optimizes a worst-case risk under shift-interventions on anchors up to a given strength. This in turn increases the robustness of the predictions to distribution shifts at the cost of reducing the in-distribution generalization.

In the finite-sample case with $n$ observations from $P_{\text{train}}$, let matrix $\mathbf{X} \in \mathbb{R}^{n \times d}$ contain the observations of $X$ and let $\mathbf{Y} \in \mathbb{R}^n$ be the vector of corresponding targets. Similarly, we denote the matrix containing the observations of $A$ with $\mathbf{A} \in \mathbb{R}^{n \times q}$ and we use $\mathbf{\Pi_A} = \mathbf{A} \left( \mathbf{A}^T \mathbf{A} \right)^\dagger \mathbf{A}^T$ as the projection operator on the column space of the anchor matrix $\mathbf{A}$ where $\mathbf{A}^\dagger$ denotes the pseudo-inverse of matrix $\mathbf{A}$. Further, $\mathbf{I}$ denotes the identity matrix. Then, the finite-sample optimization regression problem can be written as

$$\hat{b}_{\gamma, \mathbf{A}} = \arg\min_b \|(\mathbf{I} - \mathbf{\Pi_A})(\mathbf{Y} - \mathbf{X}b)\|_2^2 + \gamma \|\mathbf{\Pi_A}(\mathbf{Y} - \mathbf{X}b)\|_2^2. \tag{4}$$

The AR regression estimate $\hat{b}_{\gamma, \mathbf{A}}$ can be obtained by applying the OLS solution to a modified set of inputs and outputs:

$$\tilde{\mathbf{X}}_{\gamma, \mathbf{A}} = \mathbf{X} + (\sqrt{\gamma} - 1)\mathbf{\Pi_A}\mathbf{X} \tag{5}$$

$$\tilde{\mathbf{Y}}_{\gamma, \mathbf{A}} = \mathbf{Y} + (\sqrt{\gamma} - 1)\mathbf{\Pi_A}\mathbf{Y} \tag{6}$$

## 3 Anchor Data Augmentation

In this section, we introduce Anchor Data Augmentation (ADA), a domain-independent data augmentation method inspired by AR. ADA does not require previous knowledge about the data invariances nor manually engineered transformations. As opposed to existing domain-agnostic data augmentation methods [10, 45, 46], we do not require training of an expensive generative model, and the augmentation only adds marginally to the computation complexity of the training. In addition, since ADA originates from a causal regression problem, it can be readily applied to regression problems. Even when ADA does not improve performance, its effect on performance remains minimal.

Data augmentation aims to introduce informative data in addition to the original dataset during the training procedure of the model to improve its generalization. Similar to AR, ADA employs a linear projection, given by the anchor variables $A$, to determine the most relevant perturbation directions based on the similarity of the samples. ADA inherits the generalization properties from AR. In [42], the authors recommend that the anchor variable can be set as *an indicator of the datasets, where each dataset is a homogeneous set of observations*. A key insight of our work is that this can be achieved by clustering the data into $q$ clusters. The matrix $\mathbf{A} \in \mathbb{R}^{n \times q}$ is then constructed as an indicator matrix with a one-hot encoding of the assigned cluster index per row. For our experiments, we use *k-means* clustering [35] to construct $\mathbf{A}$. Further, in AR, only one value for $\gamma$ is used, which should be chosen based on the desired strength of perturbations on test datasets, in comparison to the training dataset [42]. We suggest that the value of $\gamma$ is sampled from a distribution with density $p(\gamma)$. In our experiments, we use a uniform distribution between $1/\alpha$ and $\alpha$, where $\alpha > 1$ is a hyperparameter to be tuned.

ADA augments a sample $(\mathbf{X}^{(i)}, \mathbf{Y}^{(i)})$ by normalizing the original AR modifications (5 and 6) by $1 + (\sqrt{\gamma} - 1) \sum_j (\mathbf{\Pi_A})^{(ij)}$ to unify the noise level across the augmentations independent of the value of $\gamma$, while approximately preserving the potentially nonlinear relation between $X$ and $y$ (see also section 3.2):

$$\tilde{\mathbf{X}}_{\gamma, \mathbf{A}}^{(i)} = \frac{\mathbf{X}^{(i)} + (\sqrt{\gamma} - 1)(\mathbf{\Pi_A})^{(i)}\mathbf{X}}{1 + (\sqrt{\gamma} - 1) \sum_j (\mathbf{\Pi_A})^{(ij)}}, \tag{7}$$

$$\tilde{\mathbf{Y}}_{\gamma, \mathbf{A}}^{(i)} = \frac{\mathbf{Y}^{(i)} + (\sqrt{\gamma} - 1)(\mathbf{\Pi_A})^{(i)}\mathbf{Y}}{1 + (\sqrt{\gamma} - 1) \sum_j (\mathbf{\Pi_A})^{(ij)}}, \tag{8}$$

where we denote $(\mathbf{M})^{(i)}$, $(\mathbf{M})^{(:j)}$, and $(\mathbf{M})^{(ij)}$ denote respectively the $i$-th row, the $j$-th column and $(i, j)$ component of some matrix $\mathbf{M}$. As is standard practice, we rely on stochastic gradient descent to optimize our (nonlinear) regressors and apply ADA on each minibatch rather than the entire dataset.

ADA combines samples from the same cluster and generates augmented samples along the data manifold. For a general $\mathbf{A}$, $\mathbf{\Pi_A}$ provides a "collective" mixing approach for the samples in a batch

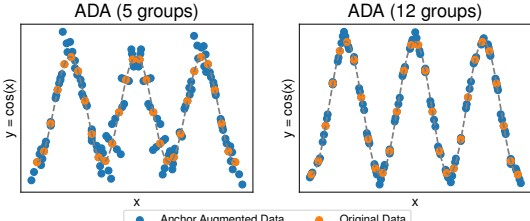

Figure 1: Comparison of ADA augmentations on a nonlinear Cosine data model. For a larger partition size, ADA augmentations are more accurate due to the high local variability of the Cosine function. We used *k-means* clustering to construct $\mathbf{A}$ and $\gamma \in \{1/2, 2/3, 1.03/2, 2.0\}$.

by determining a center, while $\gamma$ controls the extend of contraction or expansion of the augmented sample around this center. In particular, for a one-hot encoding matrix $\mathbf{A}$, $\mathbf{\Pi_A}^{(i)}\mathbf{X}$ defines the centroid of the cluster to which sample $i$ belongs. Then, the modified samples are located on the ray that originates from the centroid and goes through the original data point $(\mathbf{X}^{(i)}, \mathbf{Y}^{(i)})$. As $\gamma$ increases, the augmented samples move towards their corresponding centroid and specifically, for $\gamma = 1$ they coincide with the original samples. Furthermore, the cluster size, regulated by the number of clusters $q$, directly impacts the number of samples mixed together; with smaller clusters, fewer samples are combined. Applying ADA on each minibatch introduces further diversity and enhances robustness, because the composition of samples being mixed together and the value of $\gamma$ changes in each minibatch. In Appendix A.2 we provide a detailed explanation and analysis of the impact of ADA hyperparameters, $q$ controlling the number of clusters and $\alpha$ controlling the range of values for $\gamma$. In Appendix B.4 we empirically show how regression performance varies with respect to these hyperparameters.

In Figure 1, we visually illustrate the augmentation effects of ADA. We uniformly sampled 30 data points between $\pm3$ (i.e. $x_i \sim \mathcal{U}[-3, 3]$) and set the corresponding target variable as $y_i = \cos(\pi x_i)$ without added noise. We then clustered this data in $q = 5$ and $q = 12$ groups using k-means and applied eq. (7) and eq. (8) to the 30 samples with $\gamma \in \{1/2, 2/3, 1, 3/2, 2\}$ resulting in 150 augmented data points.

## 3.1 Comparison to C-Mixup

ADA can be interpreted as a generalized variant of C-Mixup [49]. In C-Mixup samples are mixed in pairs, and the combination probability of each sample pair is given by the similarity of their labels, measured by a Gaussian kernel. Augmented samples are then obtained as the convex combination between the pair. In contrast, ADA allows mixing multiple samples based on their cluster membership and the resulting augmentations that may reside in the convex hull of the original samples of a cluster if $\gamma \geq 1$ or beyond it when $\gamma < 1$. In particular, ADA and C-Mixup augmentations would be similar if the anchor matrix $\mathbf{A}$ indicates pairs of samples weighted by the similarity of their labels and $\gamma > 1$.

## 3.2 Preserving nonlinear data structure

In the following we show that the scaled transformations in eq. (5)) and eq. (6) preserve the nonlinear relationship, so that we can use the modified pair $(\tilde{\mathbf{X}}_{\gamma,\mathbf{A}}, \tilde{\mathbf{Y}}_{\gamma,\mathbf{A}})$ to augment the dataset $(\mathbf{X}, \mathbf{Y})$. Let $(\mathbf{X}^{(i)}, \mathbf{Y}^{(i)})$ be the $i$th sample from $P_{\text{train}}$ corresponding to the $i$th row of $\mathbf{X}$ and $i$th component of $\mathbf{Y}$. When the data has a nonlinear relation,

$$\mathbf{Y}^{(i)} = f_b(\mathbf{X}^{(i)}) + \epsilon^{(i)} \tag{9}$$

given the zero mean noise variable $\epsilon^{(i)}$, we can alter the anchor loss accordingly [4],

$$b_{\text{NONLIN},\gamma,A}, f_{\gamma,A} = \arg\min_{b,f} \mathbb{E}_{P_{\text{train}}}[((\text{Id} - \mathbf{P}_A)(y - f_b(X)))^2] + \gamma\mathbb{E}_{P_{\text{train}}}[(\mathbf{P}_A(y - f_b(X)))^2],$$

The AR modification Equations 5 and 6 do not preserve the nonlinear relation between the target and predictors,

$$\tilde{\mathbf{Y}}^{(i)} \neq f_b(\tilde{\mathbf{X}}^{(i)}) + \tilde{\epsilon}^{(i)}$$

with another zero mean variable $\tilde{\epsilon}^{(i)}$ operating as the observation noise in the augmented data. Therefore, we propose to further extend the original AR and perform the data augmentation with

scaled transformations to get the modified sample $(\tilde{\mathbf{X}}^{(i)}_{\gamma,\mathbf{A}}, \tilde{\mathbf{Y}}^{(i)}_{\gamma,\mathbf{A}})$ which approximately preserves the nonlinear relationship of sample $(\mathbf{X}^{(i)}, \mathbf{Y}^{(i)})$ as shown below.

We can rewrite $\tilde{\mathbf{Y}}^{(i)}_{\gamma,\mathbf{A}}$ in Equation (8) as

$$\tilde{\mathbf{Y}}^{(i)}_{\gamma,\mathbf{A}} = \frac{f_b(\mathbf{X}^{(\mathbf{i})}) + (\sqrt{\gamma}-1)(\mathbf{\Pi_A})^{(i)}\mathbf{F}_b(\mathbf{X})}{1 + (\sqrt{\gamma}-1)\sum_j(\mathbf{\Pi_A})^{(ij)}} + \tilde{\epsilon}^{(i)}_{\gamma,\mathbf{A}}$$

where $\tilde{\epsilon}^{(i)}_{\gamma,\mathbf{A}}$ is a zero mean noise variable and $\mathbf{F}_b(\mathbf{X}) = [f_b(\mathbf{X}^{(1)}), ..., f_b(\mathbf{X}^{(n)})]^T$. In Appendix A.1, for continuously differentiable function $f$, we can use the first order Taylor expansion of $\tilde{\mathbf{Y}}^{(i)}_{\gamma,\mathbf{A}}$ around $\tilde{\mathbf{X}}^{(i)}_{\gamma,\mathbf{A}}$ to show that

$$\tilde{\mathbf{Y}}^{(i)}_{\gamma,\mathbf{A}} \approx f_b(\tilde{\mathbf{X}}^{(i)}_{\gamma,\mathbf{A}}) + \tilde{\epsilon}^{(i)}_{\gamma,\mathbf{A}} \tag{10}$$

which approximately has the same nonlinear relation as the original model for small $\|\mathbf{X}^{(\mathbf{i})} - \tilde{\mathbf{X}}^{(i)}_{\gamma,\mathbf{A}}\|_2$ or $\|\sum_j(\mathbf{\Pi_A})^{(ij)}(\mathbf{X}^{(j)} - \tilde{\mathbf{X}}^{(i)}_{\gamma,\mathbf{A}})\|_2$.

With the one-hot partitioning matrix, $\mathbf{A}$ (introduced in the previous section), the approximation of the true nonlinear model becomes accurate in partitions with small diameter (where we define partition diameter as the maximum distance of two samples $\mathbf{X}^{(i)}$ and $\mathbf{X}^{(j)}$ in the same cluster).

### 3.3 Algorithm

Finally, in this section, we present the ADA algorithm step by step (Algorithm 1) to generate minibatches of data that can be used to train neural networks (or any other nonlinear regressor) by any stochastic gradient descent method. As discussed previously, we propose to repeat the augmentation with different parameter combinations for each minibatch.

Given a centered training dataset $(\mathbf{X}, \mathbf{Y})$, its clustering assignment $\mathbf{A}$, and prior function $p(\gamma)$, the ADA minibatch algorithms takes $L$ random samples from the training set and its corresponding rows in $\mathbf{A}$ and outputs an $L$-sample mini-bath $(\tilde{\mathbf{X}}_{\gamma,\mathbf{A}}, \tilde{\mathbf{Y}}_{\gamma,\mathbf{A}})$.

In order to do so, we first choose $\gamma$ according to the provided criterion $p(\gamma)$ (line 3). The corresponding projection matrix $\mathbf{\Pi}_A$ is computed from $\mathbf{A}$ (line 4). Finally, in lines five to seven, the transformation is applied according to Equations 7 and 8.

---

**Algorithm 1** ADA: Minibatch generation

---

1: **Input:** $L$ training data points $(\mathbf{X}, \boldsymbol{Y})$;
   prior distribution for $\gamma$: $p(\gamma)$
   $L \times q$ binary matrix $\mathbf{A}$ with a one per row indicating the clustering assignment for each sample.
2: **Output:** $(\tilde{\mathbf{X}}, \tilde{\boldsymbol{Y}})$
3: Sample $\gamma$ from $p_{(\gamma)}$
4: Projection matrix: $\mathbf{\Pi}_A \leftarrow \mathbf{A}(\mathbf{A}^T\mathbf{A})^\dagger\mathbf{A}^T$
5: **for** $i = 0$ **to** row of $\mathbf{X}$ **do**
6: $\quad \tilde{\mathbf{X}}^{(i)}_{\gamma,\mathbf{A}} \leftarrow \frac{\mathbf{X}^{(i)}+(\sqrt{\gamma}-1)(\mathbf{\Pi_A})^{(i)}\mathbf{X}}{1+(\sqrt{\gamma}-1)\sum_j(\mathbf{\Pi_A})^{(ij)}}$
7: $\quad \tilde{\mathbf{Y}}^{(i)}_{\gamma,\mathbf{A}} \leftarrow \frac{\mathbf{Y}^{(i)}+(\sqrt{\gamma}-1)(\mathbf{\Pi_A})^{(i)}\mathbf{Y}}{1+(\sqrt{\gamma}-1)\sum_j(\mathbf{\Pi_A})^{(ij)}}$
8: **end for**
9: **return** $(\tilde{\mathbf{X}}_{\gamma,\mathbf{A}}, \tilde{\boldsymbol{Y}}_{\gamma,\mathbf{A}})$

---

## 4 Experiments

We experimentally investigate and compare the performance of ADA. First, we use ADA in an in-distribution setting for a linear regression problem (Section 4.1), in which we show that even in this case, ADA provides improved performance in the low data regime. Second, in Section 4.2, we apply ADA and C-Mixup to the California and Boston Housing datasets as we increase the number of training samples. In the last two subsections, we replicate the in-distribution generalization (Section 4.3) and the out-of-distribution Robustness (Section 4.4) from the C-Mixup paper [49]. In [49] the authors further assess a task generalization experiment. However, the corresponding code was not publicly provided, and a comparison could not be easily made.

## 4.1 Linear synthetic data

Using synthetic linear data, we investigate if ADA can improve model performance in an over-parameterized setting compared to C-Mixup, vanilla augmentation, or classical expected risk mini-mization (ERM). Additionally, we analyze the sensitivity of our approach to the choice of $\gamma$ and the number of augmentations.

**Data:** The generated data follows a standard linear structure

$$\mathbf{Y}^{(i)} = \left(\mathbf{X}^{(i)}\right)^T b + b_0 + \epsilon^{(i)} \tag{11}$$

with $\mathbf{X}^{(i)}, b \in \mathbb{R}^{19}$ and $\mathbf{Y}^{(i)}, b_0, \epsilon^{(i)} \in \mathbb{R}$. The parameters are sampled randomly from a Gaussian distribution $N(0,1)$. We sample 20 different training datasets and one validation set with $\epsilon \sim \mathcal{N}(0, 0.1^2)$, $\mathbf{X}^{(i)} \sim \mathcal{N}(\mathbf{0}, \mathbf{I}_{19})$. For each training set, we take subsets with an increasing number of samples to evaluate the methods on different levels of data availability. The subsets are hierarchically constructed (i.e., meaning a smaller set is always a subset of a larger one). The validation set has 100,000 samples.

**Models and Comparisons:** We investigate and compare the impact of ADA using two different models with varying complexity: a linear Ridge regression and a multilayer perceptron (MLP) with one hidden layer and 10 units with ReLU activation. Using an MLP with more hidden layers shows similar results (see Appendix B.1 for details).

The ERM models only use the original data. We perform vanilla data augmentation by adding Gaussian noise $\epsilon' \sim N(0, 0.1^2)$ to the output leaving the input unchanged. Next, we apply C-Mixup with a bandwidth of 1 and set the $\alpha$ of the Beta-distribution to 2. Finally, we apply ADA with varying the number of obtained augmentations $k = \{10, 100\}$ and varying range of values for $\gamma$. To be precise, we define $\alpha \in \{2, 4, 6, 8, 10\}$ and specify $\beta_i = 1 + \frac{\alpha - 1}{k/2} \cdot i$ (with $i \in \{1, ..., k/2\}$) and $\gamma \in \{\frac{1}{\alpha}, \frac{1}{\beta_{k/2-1}}, ..., \frac{1}{\beta_1}, 1, \beta_1, ..., \beta_{k/2-1}, \alpha\}$. $\mathbf{A}$ is constructed using k-means clustering with $q = 8$.

For the Ridge regression model, we increase the dataset by a factor of 10 by sampling from the respective augmentation methods and subsequently compute the regression estimators. In contrast, for the MLP, we implement the augmentation methods on a minibatch level. Specifically, we incorporate vanilla augmentation by adding Gaussian noise to each batch, apply C-Mixup after sampling from the beta distribution in each batch, and finally, apply ADA after sampling from the defined gamma values in each batch.

**Results:** We plot our results in Figure 2. First, as expected, Ridge regression outperforms the MLP model. Second, when there is little data availability, using ADA decreases the test error compared to ERM. The effect diminishes when the training dataset is sufficiently large, and all models converge to the noise limit of $0.1^2$. Third, vanilla augmentation achieves similar results as ADA and C-Mixup for Ridge regression but not quite for the MLP. This suggests that ADA (and C-Mixup) are more meaningful than randomly adding noise and especially well suited for highly parameterized models as the MLP has almost 20 times more parameters than Ridge regression. In real-world applications, the value of $\epsilon$ is usually unknown, and choosing $\epsilon'$ for vanilla augmentation is not trivial, especially when the number of samples is small. Fourth, we conclude that generating more augmentations (100 instead of 10) further improves prediction error in vanilla and anchor augmentation (Appendix B Figure 9) and the effectiveness of anchor augmentation is further increased as the range for $\gamma$ is wider (Appendix B Figure 10). Finally, C-Mixup and ADA perform similarly with ADA having a tendency to achieve a lower test error.

In summary, even in the simplest of cases, in which we should not expect gains from ADA (or C-Mixup), these data augmentation strategies provide gains in performance when the number of training examples is not sufficient to achieve the error floor.

## 4.2 Housing nonlinear regression

We extend the results from the previous section to the California and Boston housing data and compare ADA to C-Mixup [49]. We repeat the same experiments on three different regression datasets. Results are provided in Appendix B.2 and also show the superiority of ADA over C-Mixup for data augmentation in the implemented experimental setup.

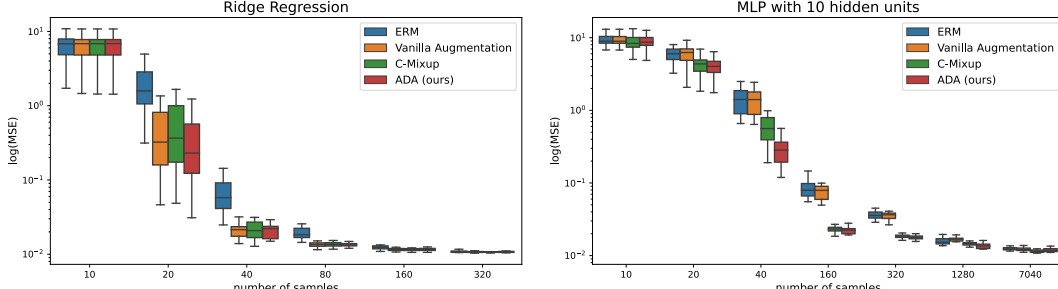

Figure 2: Mean Squared Error for Ridge Regression model and MLP model with varying number of training samples. For Ridge regression, vanilla augmentation and C-Mixup generate $k = 10$ augmented observations per observations. Similarly, Anchor Augmentation generates $k = 10$ augmented observations per observation with parameter $\alpha = 10$.

**Data:** We use the California housing dataset [19] and the Boston housing dataset [14]. The training dataset contains up to $n = 406$ samples, and the remaining samples are for validation. We report the results as a function of the number of training points.

**Models and comparisons:** We fit a ridge regression model (baseline) and train a MLP with one hidden layer with a varying number of hidden units with sigmoid activation. The baseline models only use only the original data. We train the same models using C-Mixup with a Gaussian kernel and bandwidth of 1.75. We compare the previous approaches to models fitted on ADA augmented data. We generate 20 different augmentations per original observation using different values for $\gamma$ controlled via $\alpha = 4$ similar to what was described in Section 4.1. The Anchor matrix is constructed using k-means clustering with $q = 10$.

**Results:** We report the results in Figure 3. First, we observe that the MLPs outperform Ridge regression suggesting a nonlinear data structure. Second, when the number of training samples is low, applying ADA improves the performance of all models compared to C-Mixup and the baseline. The performance gap decreases as the number of samples increases. When comparing C-Mixup and ADA, we see that using sufficiently many samples both methods achieve similar performance. While on the Boston data, the performance gap between the baseline and ADA persists, on California housing, the non-augmented model fit performs better than the augmented one when data availability increases. This suggests that there is a sweet spot where the addition of original data samples is required for better generalization, and augmented samples cannot contribute any further.

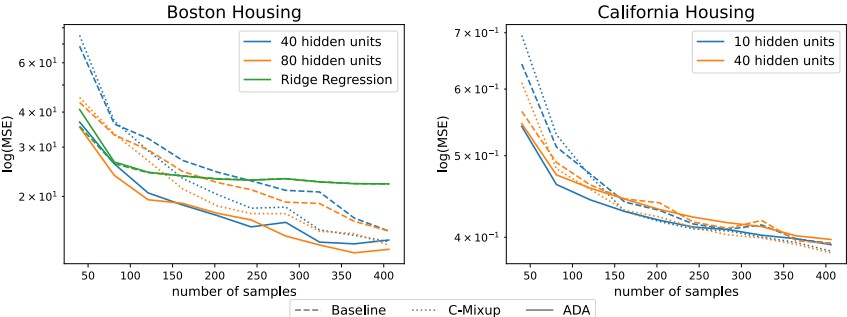

Figure 3: MSE for housing datasets averaged over 10 different train-validation-test splits. On California housing Ridge regression performs much worse which is why it is not considered further (see Appendix B.2).

### 4.3 In-distribution Generalization

In this section, we evaluate the performance of ADA and compare it to prior approaches on tasks involving in-distribution generalization. We use the same datasets as [49] and closely follow their experimental setup.

**Data:** We use four of the five in-distribution datasets used in [49]. The validation and test data are expected to follow the same distribution as the training data. Airfoil Self-Noise (Airfoil) and NO2 [24] are both tabular datasets, whereas Exchange-Rate and Electricity [27] are time series datasets. We divide the datasets into train-, validation- and test data randomly, as the authors of C-Mixup did. For Echocardiogram videos [37] (the 5th dataset in [49]), we could not replicate their preprocessing.

**Models and comparisons:** We compare our approach, ADA, to C-Mixup [49], Local-Mixup [2], Manifold-Mixup [46], Mixup [51] and classical expected risk minimization (ERM). Following the work of [49], we use the same model architectures: a three-layer fully connected network for the tabular datasets; and an LST-Attn [27] for the time series.

We follow the setup of [49] and apply C-Mixup, Manifold-Mixup, Mixup, and ERM with their reported hyperparameters and provided code. For the ADA and Local-Mixup experiments, we use hyperparameter tuning and grid search to find the optimal training (batch size, learning rate, and number of epochs), and Local-Mixup parameters (distance threshold $\epsilon$) and ADA parameters (number of clusters, range of $\gamma$, and whether to use manifold augmentation). We provide a detailed description in Appendix B.4. The evaluation metrics are Root Mean Squared Error (RMSE) and Mean Averaged Percentage Error (MAPE).

**Results:** We report the results in Table 1. For full transparency, in the last row, we copy the results from [49]. We can assess that ADA is competitive with C-Mixup and superior to the other data augmentation strategies. ADA consistently improves the regression fit compared to ERM. Under the same conditions (split of data and Neural network structure), ADA is superior to C-Mixup. But, the degree of improvement is marginal on some datasets and as we show in the last row, we could not fully replicate their results. The only data in which ADA is significantly better than C-Mixup and the other strategies is for the Airfoil data, in which ADA reduces the error by around 15% with respect to the ERM solution.

Table 1: Results for in-distribution generalization. We report the average RMSE and MAPE of three different seeds. Standard deviations are reported in Appendix B.4. The best results per column are printed in bold and the second-best results are underlined (not applicable to the last row).

| | Airfoil | | NO2 | | Exchange-Rate | | Electricity | |
|---|---|---|---|---|---|---|---|---|
| | RMSE | MAPE | RMSE | MAPE | RMSE | MAPE | RMSE | MAPE |
| **ERM** | 2.758 | 1.694 | 0.529 | 13.402 | 0.024 | 2.437 | 0.058 | 13.915 |
| **Mixup** | 3.264 | 1.964 | 0.522 | 13.226 | 0.025 | 2.513 | 0.058 | 13.839 |
| **ManiMixup** | 3.092 | 1.871 | 0.528 | 13.358 | 0.025 | 2.541 | 0.058 | 14.031 |
| **Local Mixup** | 3.373 | 2.043 | 0.524 | 13.309 | 0.021 | 2.136 | 0.063 | 14.238 |
| **C-Mixup** | 2.800 | 1.629 | 0.516 | **13.069** | 0.024 | 2.431 | **0.057** | 13.512 |
| **ADA** | **2.360** | **1.373** | **0.515** | 13.128 | **0.021** | **2.116** | 0.059 | **13.464** |
| **C-Mixup** in [49] | 2.717 | 1.610 | 0.509 | 12.998 | 0.020 | 2.041 | 0.057 | 13.372 |

## 4.4 Out-of-distribution Robustness

In this section, we evaluate the performance of ADA and compare it to prior approaches on tasks involving out-of-distribution robustness. We use the same datasets as [49] and closely follow their experimental setup.

**Data:** We use four of the five out-of-distribution datasets used in [49]. First, we use RCFashion-MNIST (RCF-MNIST) [49], which is a synthetic modification of Fashion-MNIST that models subpopulation shifts. Second, we investigate domain shifts using Communities and Crime (Crime) [12], SkillCraft1 Master Table (SkillCraft) [12] and Drug-target Interactions (DTI) [17] all of which are tabular datasets. For Crime, we use state identification, in SkillCraft we use "League Index", which corresponds to different levels of competitors, and in DTI we use year, as domain information. We split the datasets into train-, validation- and test data based on the domain information resulting in domain-distinct datasets. We provide a detailed description of datasets in Appendix B.4. Due to computational complexity, we could not establish a fair comparison on the satellite image regression dataset [23] (the fifth dataset in [49]), so we report some exploratory results in Appendix B.4.

**Models and comparisons:** As detailed in Section 4.3. Additionally, we use a ResNet-18 [15] for RCF-MNIST and DeepDTA [38] for DTI, as proposed in [49].

**Results:** We report the RMSE and the "worst" domain RMSE, which corresponds to the worst within-domain RMSE for out-of-domain test sets in Table 2. Similar to [49], we report the $R$ value for the DTI dataset (higher values suggest a better fit of the regression model). For full transparency, in the last row, we copy the results from [49]. We can assess that ADA is competitive with C-Mixup and the other data augmentation strategies. Under the same conditions (split of data and Neural network structure), ADA is superior to C-Mixup. But, the degree of improvement is marginal on some datasets and as we show in the last row, we could not fully replicate their results. ADA is significantly better than C-Mixup and other strategies on the SkillCraft data, in which ADA reduces the error by around $15\%$ compared to the ERM solution.

Table 2: Results for out-of-distribution generalisation. We report the average RMSE across domains in the test data and the "worst within-domain RMSE over three different seeds. For the DTI dataset, we report average R and "worst within-domain" R. Standard deviations are reported in Appendix B.4. The best results per column are printed in bold and the second-best results are underlined (not applicable to the last row).

| | RCF-MNIST | Crimes | | SkillCraft | | DTI | |
| | avg. RMSE | avg. RMSE | worst RMSE | avg. RMSE | worst RMSE | avg. R | worst R |
|---|---|---|---|---|---|---|---|
| **ERM** | 0.164 | 0.136 | 0.170 | 6.147 | 7.906 | 0.483 | 0.439 |
| **Mixup** | 0.159 | 0.134 | 0.168 | 6.460 | 9.834 | 0.459 | 0.424 |
| **ManiMixup** | **0.157** | **0.128** | **0.155** | 5.908 | 9.264 | 0.474 | 0.431 |
| **LocalMixup** | 0.187 | 0.133 | 0.1590 | 7.251 | 10.996 | 0.470 | 0.433 |
| **C-Mixup** | 0.158 | 0.132 | 0.165 | 6.216 | 8.223 | 0.474 | 0.435 |
| **ADA** | 0.175 | 0.130 | 0.156 | **5.301** | **6.877** | **0.493** | **0.448** |
| **C-Mixup** in [49] | 0.146 | 0.123 | 0.146 | 5.201 | 7.362 | 0.498 | 0.458 |

## 5 Conclusion

We introduced Anchor Data Augmentation (ADA), an extension of Anchor Regression for the purpose of data augmentation. AR is a novel causal approach to increase the robustness in regression problems. In ADA, we systematically mix multiple samples based on a collective similarity criterion, which is determined via clustering. The augmented samples are modifications of the original samples that are moved towards or away from the cluster centroids based on the desired degree of robustness in AR. Our empirical evaluations across diverse synthetic and real-world regression problems consistently demonstrate the effectiveness of ADA, especially for limited data availability. ADA is competitive with or outperforms state-of-the-art data augmentation strategies for regression problems, even though the improvements are marginal on some datasets.

ADA can be applied to any regression setting, and we have not found any case in which the results were detrimental. To apply ADA, we only need to cluster our data and select a distribution for $\gamma$. We relied on vanilla *k-means*, and the results are robust with respect to the number of clusters. Other clustering algorithms might be more suitable for different applications. For setting the parameter $\gamma$, we used a uniform distribution. We believe a gamma distribution could be equally effective.

## Broader Impact

The purpose of data augmentation is to compensate for data scarcity in multiple domains where gathering and labeling data accurately by experts is impractical, expensive, or time-consuming. If applied properly, it can effectively expand the training dataset, reduce overfitting and improve the model's robustness, as was shown in the paper. However, It is important to note that the choice and combination of the data augmentation technique depends on the specific problem and using the wrong augmentation method may introduce additional bias to the model. More generally, incorrect data augmentation can lead to the following problems: overfitting the augmented data, loss of important information, introduction of unrealistic patterns and imbalanced presentation of the data. Detecting emerging problems due to data augmentation may not be straightforward. In particular, the performance on a test distribution that matches the training data distribution may be misleading and the model's predictions should be used with caution on new data that reflects the potential distribution shifts or variations encountered in real-world.

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
