# A Additional information for Anchor Data Augmentation

## A.1 Derivation of ADA for nonlinear data

In the following, we provide the more detailed derivation to Equation (10), which motivates the usage of the scaled transformation we use in ADA to obtain $(\tilde{\mathbf{X}}_{\gamma,\mathbf{A}}, \tilde{\mathbf{Y}}_{\gamma,\mathbf{A}})$. We use the same notation that was introduced in Section 3. As discussed in Section 3, we can write $\tilde{\mathbf{Y}}_{\gamma,\mathbf{A}}^{(i)}$ in Equation 8 as

$$(12)$$

for some noise variable $\tilde{\epsilon}_{\gamma,\mathbf{A}}^{(i)}$, where $\mathbf{F}_b(\mathbf{X}) = [f_b(\mathbf{X}^{(1)}), ..., f_b(\mathbf{X}^{(n)})]^T$. For differentiable function $f$ with continuous first-order derivative $\dot{f}$, we can use Taylor expansion around $\tilde{\mathbf{X}}_{\gamma,\mathbf{A}}^{(i)}$ of the nominator and get

$$
\begin{aligned}
f_b(X^{(i)}) + (\sqrt{\gamma} - 1)(\mathbf{\Pi_A})^{(i)}\mathbf{F}_b(\mathbf{X}) = & f_b(\tilde{\mathbf{X}}_{\gamma,\mathbf{A}}^{(i)}) + (\mathbf{X}^{(i)} - \tilde{\mathbf{X}}_{\gamma,\mathbf{A}}^{(i)})^T \dot{f}_b(\tilde{\mathbf{X}}_{\gamma,\mathbf{A}}^{(i)}) \\
& + (\sqrt{\gamma} - 1)\sum_j (\mathbf{\Pi_A})^{(ij)} f_b(\tilde{\mathbf{X}}_{\gamma,\mathbf{A}}^{(i)}) \\
& + (\sqrt{\gamma} - 1)\sum_j (\mathbf{\Pi_A})^{(ij)}(\mathbf{X}^{(j)} - \tilde{\mathbf{X}}_{\gamma,\mathbf{A}}^{(i)}) \dot{f}_b(\tilde{\mathbf{X}}_{\gamma,\mathbf{A}}^{(i)}) \\
& + \mathcal{O}(\|\mathbf{X}^{(i)} - \tilde{\mathbf{X}}_{\gamma,\mathbf{A}}^{(i)}\|_2 \| \sum_j (\mathbf{\Pi_A})^{(ij)}(\mathbf{X}^{(j)} - \tilde{\mathbf{X}}_{\gamma,\mathbf{A}}^{(i)})\|_2) \\
= & \left( 1 + (\sqrt{\gamma} - 1)\sum_j (\mathbf{\Pi_A})^{(ij)} \right) f_b(\tilde{\mathbf{X}}_{\gamma,\mathbf{A}}) \\
& + \mathcal{O}(\|\mathbf{X}^{(i)} - \tilde{\mathbf{X}}_{\gamma,\mathbf{A}}^{(i)}\|_2 \| \sum_j (\mathbf{\Pi_A})^{(ij)}(\mathbf{X}^{(j)} - \tilde{\mathbf{X}}_{\gamma,\mathbf{A}}^{(i)})\|_2)
\end{aligned}
$$

where in the second equality we use the fact that coefficient of $\dot{f}_b(\tilde{\mathbf{X}}_{\gamma,\mathbf{A}}^{(i)})$ (in the second and fourth term) is zero for any $f_b$ due the definition of $\tilde{\mathbf{X}}^{(i)}$ in Equation 7 and therefore,

$$
\tilde{\mathbf{Y}}_{\gamma,\mathbf{A}}^{(i)} \approx f_b(\tilde{\mathbf{X}}_{\gamma,\mathbf{A}}^{(i)}) + \tilde{\epsilon}_{\gamma,\mathbf{A}}^{(i)}
$$

which is approximately similar to the original nonlinear model for small $\|\mathbf{X}^{(i)} - \tilde{\mathbf{X}}_{\gamma,\mathbf{A}}^{(i)}\|_2$ or $\| \sum_j (\mathbf{\Pi_A})^{(ij)}(\mathbf{X}^{(j)} - \tilde{\mathbf{X}}_{\gamma,\mathbf{A}}^{(i)})\|_2$.

## A.2 Additional information on hyperparameters of ADA

In this section, we illustrate in a simple 1D example (i.e. cosine data used in Figure 1) how changes in the hyperparameter values modify the data and affect the achieved estimation. Additionally, we show in Appendix B.4 how ADA performance on real-world data is impacted by changes in the hyperparameter values.

Having a fixed pair of $(\gamma, \mathbf{A})$ enforces the model to learn the optimal parameters for a particular trade-off between performance on $P_{\text{train}}$ and predefined interventional distributions [42]. Instead of limiting the regularization to a fixed pair of $(\gamma, \mathbf{A})$ that performs well on a previously known set of interventions, we propose to optimize the loss simultaneously over a set of $\gamma \in [0, \infty)$ and different anchor matrices. In particular, we optimize the parameters on a mixture of essentially similar distributions to $P_{\text{train}}$ simultaneously. To reduce the anchor regression's regularization effect, we propose using a combination of the following methods to exploit the data invariances and avoid conservative predictions.

**Anchor Matrices and Locality:** Anchor variable $A$ is assumed to be the exogenous variable that generates heterogeneity in the target and has an approximately linear relation with $(X, y)$ (see AR loss in Equation 3). It is recommended to choose the variable relying on expert knowledge about the features that the target has a higher dependence on or is possibly misrepresented in the dataset so that

we encourage the robustness of the trained model against this type of discrepancy. After deciding the features, one way to construct the anchor matrix $\mathbf{A}$ is to partition the dataset according to the similarity of the features, using for example binning or clustering algorithms. Then we can fill the rows of $\mathbf{A}$ with a one-hot encoding of the partition index that each sample belongs to.

We use the following nonlinear Cosine data model as a running example to demonstrate more clearly how $\mathbf{A}$ is constructed and affects the augmentation procedure.

$$\epsilon \sim \mathcal{N}(0, 0.1^2 \cdot \mathbf{I}), X \sim U(-3\pi, 3\pi), y = \cos(X^T b) + \epsilon, \tag{13}$$

For illustration purposes, we use in Figures 5, 7 equidistant $x$ values as this reduces noise and emphasizes the effect of ADA parameters more.

Further, we note $a : \mathcal{X} \to \{1, ..., q\}$ that maps each sample $X \in \mathcal{X}$ to one of $q$ partitions and returns its index. For instance, with an equal width binning scheme one can partition the range of a feature map $g_k : \mathcal{X} \to [0, B]$ to $q$ parts and set $a(X) := \arg\min_{r \in \{1,...,q\}} \{r : r/q \geq g_k(X)\}$. Using, equal size binning scheme, one would first sort $g_k(X^{(i)})$ for $i \in \{1, ..., n\}$ get the indices $o(g_k(X^{(i)}))$ accordingly and use $a(X^{(i)}) := \arg\min_{r \in \{1,...,q\}} \{r : rn/q \geq o(g_k(X))\}$. Similarly, it is possible to use a clustering algorithm such as k-means [35] to partition $\{X^{(i)}\}_i$ into hard clusters based on the similarity of each sample to cluster center $c_r \in \mathcal{X}$ for $r \in \{1, ..., q\}$ leading to $a(X) := \arg\min_{r \in \{1,...,q\}} D(X, c_r)$ for some distance metric $D : \mathcal{X} \times \mathcal{X} \to [0, \infty)$.

With $\mathbf{A}$ constructed from the one-hot encoding of partition indices of samples, the $\mathbf{\Pi_A}$ operator returns the average value of the projected values in the same group as each sample.

$$(\mathbf{\Pi_A})^{(i)} \mathbf{X} = \frac{1}{n_r} \sum_{j:a(\mathbf{X}^{(j)})=a(\mathbf{X}^{(i)})} \mathbf{X}^{(j)} \text{ and}$$

$$(\mathbf{\Pi_A})^{(i)} \mathbf{Y} = \frac{1}{n_r} \sum_{j:a(\mathbf{X}^{(j)})=a(\mathbf{X}^{(i)})} \mathbf{Y}^{(j)},$$

where $n_r$ is the size of group with index $r = a(\mathbf{X}^{(i)})$. Getting weighted averages of partition samples is also straightforward by scaling the one-hot encodings of group indices with the squared root of the desired weights.

**Partition Size and Number:** As was mentioned before, the target should have a high dependence on the anchor variable $A$. Specifically, with the partitioning scheme explained above, $\tilde{\mathbf{X}}_{\gamma,\mathbf{A}}^{(i)}$ is constructed as a linear combination of $\mathbf{X}^{(i)}$ and the partition average with a target variable constructed in a similar manner. If the generative function $f$ varies significantly in a partition, the average value is going to flatten out the variations and decrease the heterogeneity of the augmented samples in that partition. For a smaller partition size, the augmented data is going to be close to the mean value $f$ and improve the optimization, however, partitions with a smaller number of samples will have a noisier estimation of the sample mean in each partition and deem the augmentation ineffective. We show the same effect of $q$ on the Cosine data model in Figure 4 for $\gamma$ set via $\alpha = 2$ (as described in Section 4.1) and a different number of groups when $g_k(X) = X$ and K-Means is used for partitioning the dataset. In the groups where $f$ is approximately linear, the augmentation line is approximately tangent to $f$, specifically when the clusters are small and the cluster average lies close to $\cos(X)$.

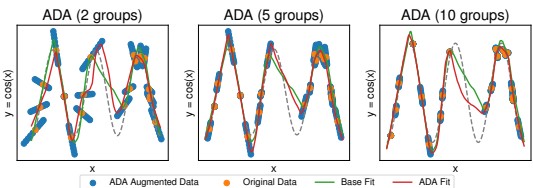

Figure 4: Model predictions for models fit on the original data and ADA augmented data with varying partition sizes. On a hold-out validation set the base model has $MSE = 0.097$. The augmented model achieves MSEs of $0.124, 0.069, 0.079$, respectively. We use MLPs with architecture $[50, 50, 50, 50, 50]$ and ReLU activation function. The original data has $n = 20$ points. We use k-means clustering, $\alpha = 2$, and augmented 10 additional points per given point.

**Values of $\gamma$:** For $\gamma \in [0, \infty)$, the transformations in Equation 7 and 8 defines a line passing through $(\mathbf{X}^{(i)}, \mathbf{Y}^{(i)})$ and the group average $((\mathbf{\Pi_A})^{(i)}\mathbf{X}, (\mathbf{\Pi_A})^{(i)}\mathbf{Y})$. As $|\gamma - 1|$ grows the augmented sample gets further away from $\mathbf{X}^{(i)}$ and in large groups this may result in misleading augmentation. Therefore, when group diameter is large it is important to keep $\gamma$ close to one. In Figure 5 we show how varying $\gamma$ changes the efficacy of the augmented samples for the Cosine data model with $q = 2$ groups. To be precise, we vary the range of $\gamma$ by defining a parameter $\alpha \in \{1.5, 2, 5, 10\}$. We further specify $\beta_i = 1 + \frac{\alpha - 1}{k/2} \cdot i$ (with $i \in \{1, ..., k/2\}$) where $k$ is the number of augmentations and finally $\gamma \in \left\{\frac{1}{\alpha}, \frac{1}{\beta_{k/2-1}}, ..., \frac{1}{\beta_1}, 1, \beta_1, ..., \beta_{k/2-1}, \alpha\right\}$. Additionally, we provide a baseline and an augmented model fit in Figure 6 with different values for $\gamma$.

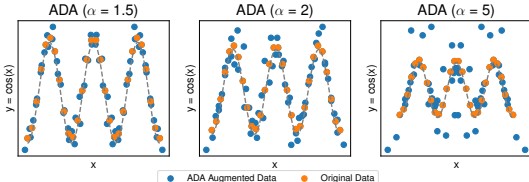

Figure 5: ADA Augmented samples for varying ranges of $\gamma$ controlled via the parameter $\alpha$. We use k-means clustering into $q = 5$ groups and augmented 2 additional points per given point.

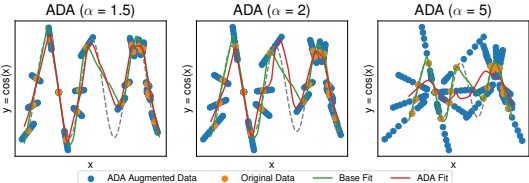

Figure 6: Model predictions for models fit on the original data and ADA augmented data with different ranges of $\gamma$ controlled via the parameter $\alpha$. On a hold-out validation set the base model has $MSE = 0.097$. The augmented model fits achieve MSEs of $0.083, 0.124, 0.470$, respectively. We use MLPs with architecture $[50, 50, 50, 50, 50]$ and ReLU activation function. The original data has $n = 20$ points. We use k-means clustering into $q = 2$ groups and augmented 10 additional points per given point.

**Number of augmentations:** For each anchor matrix $\mathbf{A}$ and $\gamma$ we can add $n$ new samples to the dataset. The addition of more augmented samples may not be beneficial as the optimization may overfit the approximations in the augmented data model in Equation 10. In the Cosine data model this is specifically problematic when $X$ is close to multiples of $\pi$ as depicted in Figure 7. Additionally, we provide a baseline and an augmented model fit in Figure 8 with different number of augmentations.

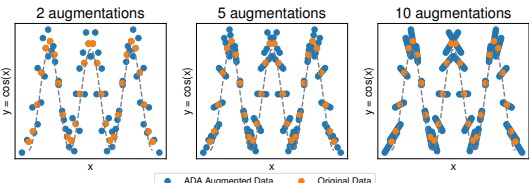

Figure 7: ADA Augmented samples for varying numbers of parameter combinations. We use k-means clustering into $q = 2$ groups $\alpha = 1.5$.

As it is standard practice to use stochastic gradient descent methods for optimizing a regressor, we suggest applying ADA on each minibatch instead of the entire dataset. This avoids choosing a fixed numbers of augmentations. Furthermore, it adds diversity to the "mixing" behavior of ADA, because the samples that are being mixed change in each iteration.

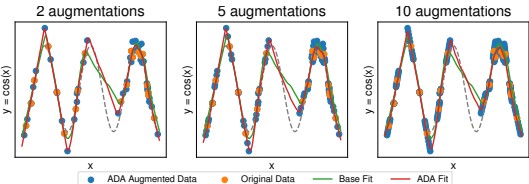

Figure 8: Model predictions for models fit on the original data and ADA augmented data with a different number of parameter combinations (equal number of augmentations). On a hold-out validation set the base model has $MSE = 0.097$. The augmented model fits achieve MSEs of $0.470, 0.071, 0.057$, respectively. We use MLPs with architecture $[50, 50, 50, 50, 50]$ and ReLU activation function. The original data has $n = 20$ points. We use k-means clustering into $q = 5$ groups and $\alpha = 2$.

# B  Experiments

## B.1  Linear synthetic data

In this section, we present more detailed results of the experiments on synthetic linear data (Section 4.1). First, Figure 9 shows a comparison of using 10 instead of 100 additional augmentations per original sample using Ridge regression model. Performance increases when using 100 instead of 10 augmentations for all methods, as the resulting prediction error is lower.

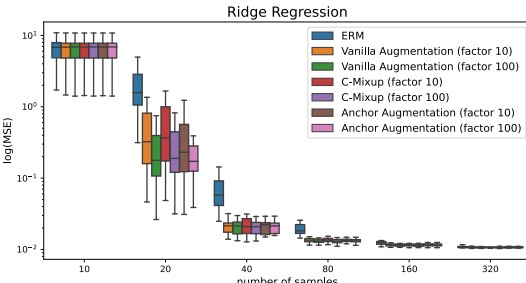

Figure 9: Comparison of augmenting the synthetic linear dataset by a factor of 10 and 100. More augmentations achieve lower MSE on all methods. Here, anchor augmentation is performed for $\alpha = 8$.

Second, we report experimental results for using a wider interval for $\gamma$ values in Figure 10. The width is controlled via the parameter $\alpha$, as described in Section 3. While for ridge regression, the effectiveness of anchor augmentation is not sensitive to the choice of $\alpha$, the MLP model shows more sensitivity.

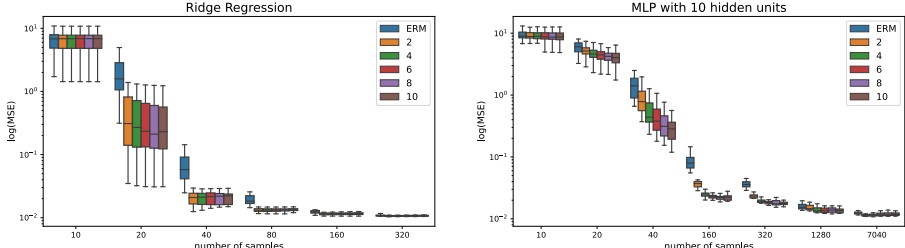

Figure 10: Comparison of augmenting the synthetic linear dataset with different intervals of $\gamma$ controlled via $\alpha$. The ridge regression is not sensitive to the choice of $\alpha$, as different values result in a similar prediction error. Contrary, for the MLP a larger value of $\alpha$ is more effective.

Finally, we report results for using an MLP with 40 hidden units in Figure 11. The results are consistent with the results from the MLP with 10 hidden unity.

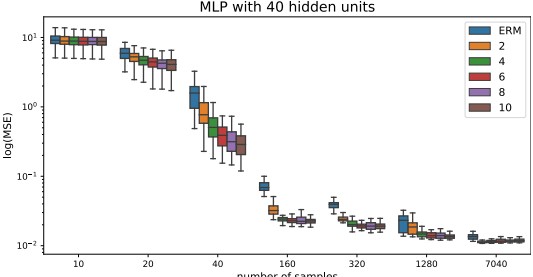

Figure 11: Analysis of the sensitivity of ADA to the choice of $\alpha$ using an MLP with 40 hidden units.

## B.2 Additional results for real-world regression data

In Figure 12 we provide additional results showing, that the Ridge regression model performs worse on the California housing data. The experimental setting is the same as described in Section 4.2.

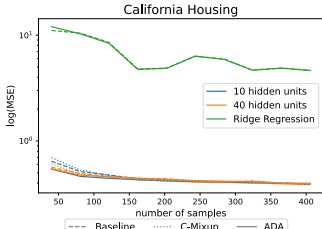

Figure 12: MSE for California housing averaged over 10 different train-validation-test splits.

## B.3 Additional results on real-world data

In this section, we provide further experimental results on real-world regression problems. We use the following datasets from the UCI ML repository [12]: *Auto MPG* (7 predictors), *Concrete Compressive Strength*(8 predictors), and *Yacht Hydrodynamics* (6 predictors). The experimental setting follows the one described in Section 4.2, except that we use here to training, validation, and test datasets of relative sizes 50%, 25%, and 25% respectively. We use MLPs with one layer and varying layer width and sigmoid activation. The models are trained using Adam optimization. We generate 9 different dataset splits and report the average prediction error in Figure 13. Similar to the results in Section 4.2, ADA outperforms the baseline and C-Mixup especially when little data is available. The performance gap vanishes when more samples are available demonstrating the effectiveness of ADA in over-parameterized scenarios.

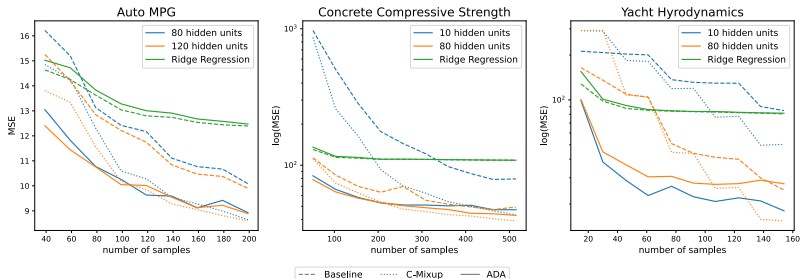

Figure 13: MSE for housing datasets averaged over 9 different train-validation-test splits.

### B.4 Details: In-distribution Generalization and Out-of-distribution Robustness

In this section we present details for the experiments described in section 4.3 and Section 4.4. We closely follow the experimental setup of [49].

**Data Description**

In the following, we provide a more detailed description of the datasets used for in-distribution generalization and out-of-distribution robustness experiments.

**Airfoil [12]:** is a tabular dataset originating from aerodynamic and acoustic tests of two and three-dimensional airfoil blade sections. Each input has 5 features measuring frequency, angle of attack, chord length, free-stream velocity and suction side displacement thickness. The target variable is the scaled sound pressure level. As in [49], we additionally apply Min-Max normalization on the input features and split the dataset into train (1003 samples), validation (300 samples) and test (200 samples) data.

**NO2:** is a tabular dataset originating from a study where air pollution at a road is related to traffic volume and meteorological variables. Each input has 7 features measuring, the logarithm of the number of cars per hour, temperature 2 meter above ground, wind speed, the temperature difference between 25 and 2 meters above ground, wind direction, hour of day and day number from 1st October 1 2001. The target variable is the logarithm of the concentration of NO2 particles, measured at Alnabru in Oslo, Norway. Following [49], we split the dataset into a train (200 samples), validation (200 samples) and test data (100 samples).

**Exchange-Rate [27]:** is a timeseries measuring the daily exchange rate of eight foreign countries including Australia, British, Canada, Switzerland, China, Japan, New Zealand and Singapore ranging from 1990 to 2016. The slide window size is 168 days, therefore the input has dimension $168 \times 8$ and the label has dimension $1 \times 8$. Following [27, 49] the dataset is split into training (4,373 samples), validation (1,518 samples) and test data (1,518 samples) in chronological order.

**Electricity [27]:** is a timeseries measuring the electricity consumption of 321 clients from 2012 to 2014. Similar to [27, 49] we converted the data to reflect hourly consumption. The slide window size is 168 hours, therefore the input has dimension $168 \times 321$ and the label has dimension $1 \times 321$. The dataset is split into training (15,591 samples), validation (5,261 samples) and test data (5,261 samples) in chronological order.

**RCF-MNIST [49]:** is rotated and colored version of F-MNIST simulating a subpopulation shift. The author rotate the images by a normalized angle $g \in [0, 1]$. In the training data they additionally color 80% of images with RGB values $[g; 0; 1 - g]$ and 20% of images with RGB values $[1 - g; 0; g]$. In the test data, they reverse the spurious correlations, so 80% of images are colored with RGB values $[1 - g; 0; g]$ and the remaining are colored with $[g; 0; 1 - g]$.

**Crime [12]:** is a tabular dataset combining socio-economic data from the 1990 US Census, law enforcement data from the 1990 US LEMAS survey, and crime data from the 1995 FBI UCR. Each input has 122 features that are supposed to have a plausible connection to crime, e.g. the median family income or per capita number of police officers. The target variable is the per capita violent crimes, representing the sum of violent crimes in the US including murder, rape, robbery, and assault. Following [49], we normalize all numerical features to the range $[0.0, 1.0]$ by equal-interval binning method and we impute the missing values using the mean value of the respective attribute. The state identifications are used as domain information. In total, there are 46 distinct domains and the data is split into disjoint domains. More precise, the training data has $1,390 samples$ with 31 domains, the validation data has 231 sampels with 6 domains and the test data has 373 samples with 9 domains.

**SkillCraft [12]:** is a tabular dataset originating from a study which uses video game from real-time strategy (RTS) games to explore the development of expertise. Each input has 17 features measuring player-related parameters, e.g. the age of the player and Hotkey usage variables. Following [49], we use the action latency in the game as a target variable. Missing values are imputed using the mean value of the respective attribute. "League Index", which corresponds to different levels of competitors, is used as domain information. In total there are 8 distinct domains and the data is split into disjoint domains. More precise, the training data has $1,878$ samples with $4$ domains, the validation data has 806 samples with 1 domain and the test data has 711 samples with 3 distinct domains.

**DTI [17]:** is a tabular dataset where the target is the binding activity score between a drug molecule and the corresponding target protein. The input consists of $32,300$ features which represent a one-hot encoding of drug and target protein information. Following [49], we use "Year" as domain information with 8 distinct domains. There are $38,400$ training, $13,440$ validation and $11,008$ test samples.

### Methods and Hyperparameters

For ERM, Mixup, ManiMixup and C-Mixup, we apply the same hyperparameters as reported in the original C-Mixup paper [49]. According to the authors they are already finetuned via a cross-validation grid search. The details can be found in the corresponding original paper. We rerun their experiments with the provided repository (https://github.com/huaxiuyao/C-Mixup) over three different seeds $\in \{0, 1, 2\}$. Furthermore, we finetune ADA and training hyperparameters using a grid search. The detailed hyperparameters for in-distribution generalization and out-of-distribution robustness are reported in Table 3. We apply ADA using the same seeds.

Table 3: Hyperparameters for ADA

|  | Airfoil | NO2 | Exchange | Electricity | RCF-MNIST | Crime | SkillCraft | DTI |
|---|---|---|---|---|---|---|---|---|
| **Architecture** | FCN3 | FCN3 | LST-Attn | LST-Att | ResNet-18 | FCN3 | FCN3 | DeepDTA |
| **Learningrate** | 0.01 | 5e-4 | 5e-4 | 5e-4 | 7e-5 | 1e-4 | 0.001 | 1e-4 |
| **Optimizer** | Adam | Adam | Adam | Adam | Adam | Adam | Adam | Adam |
| **Batchsize** | 16 | 32 | 64 | 64 | 128 | 48 | 48 | 32 |
| **Maximum Epoch** | 200 | 150 | 100 | 100 | 40 | 250 | 100 | 20 |
| $\alpha$ **(determines $\gamma$)** | 2 | 3.5 | 1.125 | 1.125 | 3 | 2.5 | 4 | 3 |
| **Number Groups** | 8 | 4 | 40 | 40 | 25 | 2 | 16 | 24 |
| **Manifold** | 1 | 0 | 0 | 0 | 1 | 1 | 0 | 1 |

Furthermore, we provide the performance of ADA for different parameter parameter values to get a better understanding of their impact. We vary values for $q$, the number of clusters used in k-means clustering, and $\alpha$, the parameter that controls the range of $\gamma$-values on selected in-distribution and out-of-distribution datasets. Results are reported in Figure 14.

Figure 14: Results for different $\alpha$ values (upper row) and $q$ values (lower row). Results are reported of three different seeds $\in \{0, 1, 2\}$. For Airfoil and Electricity we report RMSE and for Crimes we report "worst within-domain" RMSE.

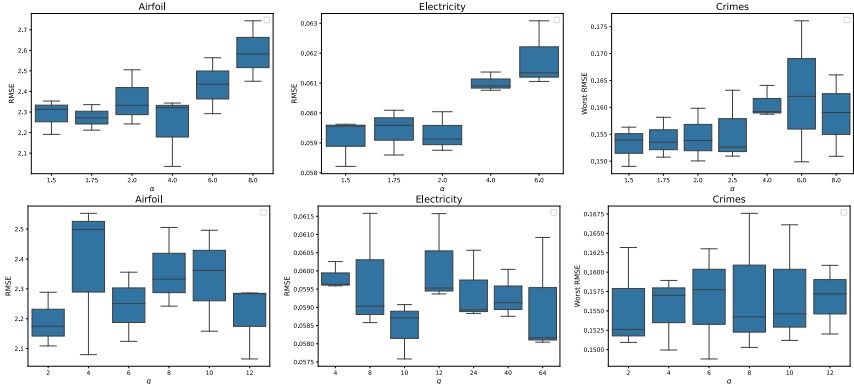

### Detailed Results

We report the results for in-distribution generalization experiments in Table 4 and for out-of-distribution generalization experiments in Table 5. Following [49], we further evaluated the performance of ADA and C-Mixup on the Poverty dataset [23], which contains satellite images from African countries and the corresponding village-level real-valued asset wealth index. Again we closely followed the experimental setup, so for details we refer to [49]. However, due to computational complexity, ADA hyperparameters are not tuned on this dataset. We use the same learningparameters as reported in [49] and $q = 24$ and $\alpha = 2$.

Table 4: Detailed results for in-distribution generalization. We report the average RMSE and MAPE and the respective standard deviations over three seeds $\in \{0, 1, 2\}$.

| | Airfoil | | | | NO2 | | | |
|---|---|---|---|---|---|---|---|---|
| | RMSE | | MAPE | | RMSE | | MAPE | |
| | mean | std | mean | std | mean | std | mean | std |
| **ERM** | 2.7582 | 0.1094 | 1.6942 | 0.0486 | 0.5294 | 0.0128 | 13.4019 | 0.3373 |
| **Mixup** | 3.2637 | 0.1633 | 1.9645 | 0.1092 | 0.5220 | 0.0037 | 13.2260 | 0.1237 |
| **ManiMixup** | 3.0922 | 0.1668 | 1.8712 | 0.0903 | 0.5277 | 0.0066 | 13.3579 | 0.2148 |
| **Local Mixup** | 3.3727 | 0.1068 | 2.0426 | 0.0211 | 0.5242 | 0.0004 | 13.3087 | 0.1990 |
| **C-Mixup** | 2.7997 | 0.2136 | 1.6289 | 0.1088 | 0.5157 | 0.0123 | 13.0688 | 0.3593 |
| **ADA** | 2.3601 | 0.1339 | 1.3730 | 0.0564 | 0.5147 | 0.0075 | 13.1277 | 0.1468 |

| | Exchange-rate | | | | Electricity | | | |
|---|---|---|---|---|---|---|---|---|
| | RMSE | | MAPE | | RMSE | | MAPE | |
| | mean | std | mean | std | mean | std | mean | std |
| **ERM** | 0.0236 | 0.0065 | 2.4366 | 0.7142 | 0.0582 | 0.0002 | 13.9153 | 0.3410 |
| **Mixup** | 0.0246 | 0.0058 | 2.5131 | 0.6667 | 0.0581 | 0.0002 | 13.8390 | 0.0539 |
| **ManiMixup** | 0.0246 | 0.0065 | 2.5411 | 0.7417 | 0.0583 | 0.0004 | 14.0308 | 0.2174 |
| **Local Mixup** | 0.0209 | 0.0046 | 2.1360 | 0.5851 | 0.0627 | 0.0054 | 14.2382 | 1.2349 |
| **C-Mixup** | 0.0238 | 0.0061 | 2.4307 | 0.6819 | 0.0573 | 0.0003 | 13.5121 | 0.0979 |
| **ADA** | 0.0209 | 0.0060 | 2.1159 | 0.6889 | 0.0587 | 0.0008 | 13.4642 | 0.2956 |

Table 5: Detailed results for out-of-distribution generalization. We report the average and standard deviation of average and worst RMSE or R over three seeds $\in \{0, 1, 2\}$.

| | RCF-MNIST | | Crime | | | | SkillCraft | | | |
|---|---|---|---|---|---|---|---|---|---|---|
| | avg. RMSE | | avg. RMSE | | worst RMSE | | avg. RMSE | | worst RMSE | |
| | mean | std | mean | std | mean | std | mean | std | mean | std |
| **ERM** | 0.1636 | 0.0066 | 0.1356 | 0.0057 | 0.1698 | 0.0066 | 6.1473 | 0.4070 | 7.9064 | 0.3223 |
| **Mixup** | 0.1585 | 0.0048 | 0.1341 | 0.0031 | 0.1681 | 0.0171 | 6.4605 | 0.4259 | 9.8338 | 0.9415 |
| **ManiMixup** | 0.1572 | 0.0205 | 0.1283 | 0.0030 | 0.1554 | 0.0086 | 5.9080 | 0.3438 | 9.2643 | 1.0117 |
| **Local Mixup** | 0.1873 | 0.0179 | 0.1325 | 0.0033 | 0.1590 | 0.0052 | 7.2514 | 0.4121 | 10.9957 | 0.5702 |
| **C-Mixup** | 0.1579 | 0.0066 | 0.1320 | 0.0017 | 0.1647 | 0.0045 | 6.2156 | 0.3822 | 8.2232 | 0.5463 |
| **ADA** | 0.1629 | 0.0142 | 0.1298 | 0.0032 | 0.1556 | 0.0066 | 5.3014 | 0.1821 | 6.8771 | 1.2666 |

| | DTI | | | | Poverty Map | | | |
|---|---|---|---|---|---|---|---|---|
| | avg. R | | worst R | | avg. R | | worst R | |
| | mean | std | mean | std | mean | std | mean | std |
| **ERM** | 0.4827 | 0.0080 | 0.4391 | 0.0154 | n/a | n/a | n/a | n/a |
| **Mixup** | 0.4589 | 0.0131 | 0.4239 | 0.0025 | n/a | n/a | n/a | n/a |
| **ManiMixup** | 0.4736 | 0.0040 | 0.4306 | 0.0087 | n/a | n/a | n/a | n/a |
| **Local Mixup** | 0.4700 | 0.0127 | 0.4325 | 0.0075 | n/a | n/a | n/a | n/a |
| **CMixup** | 0.4735 | 0.0041 | 0.4346 | 0.0082 | 0.8040 | 0.0396 | 0.5388 | 0.0725 |
| **ADA** | 0.4928 | 0.0098 | 0.4483 | 0.0094 | 0.7938 | 0.0328 | 0.5218 | 0.0616 |