# OpenReview forum: "Anchor Data Augmentation"
_NeurIPS.cc/2023/Conference — NeurIPS 2023 poster_

### Official Review · Reviewer_dYVk · 2023-07-10

**Soundness:** 3 good
**Presentation:** 3 good
**Contribution:** 3 good
**Rating:** 6
**Confidence:** 2

**Summary:**

The authors proposed a new data augmentation algorithm for regression datasets. They borrow ideas from Anchor regression model which captures the heterogeneity of the dataset using anchor variables to create additional augmented data. They show that applying this type of augmentation improves results compared to naive mixup augmentation on many datasets.

**Strengths:**

- Well-written paper.
- The augmentation proposed is theoretically motivated.
- Evaluated on many datasets and scenarios (ID, OOD datasets etc).



**Weaknesses:**

- The results are not significantly better than the baselines in low and high data regimes.
- The theoretical motivation from Anchor Regression is not clearly explained or justified.
- The comparison is limited only to C-Mixup as a prior method, could be broader.
- More analysis could be provided on why and how ADA improves robustness.
- (As far as I understand) ADA has more hyperparameters to tune compared to some other simpler data augmentation techniques.


**Questions:**

Will your model generalize to datasets with only categorical variables?
Do you have ablations on different values of "k" and "alpha"?

**Limitations:**

Yes

---

> ### Author Rebuttal · Authors · 2023-08-09
>
> Thank you for your direct review and questions about our procedure. We will address them in the following paragraphs.
>
> Data augmentation for regression is hard. As we mentioned in our paper, there are two papers on the topic. The results sometimes are much better than the baseline, but in most cases, they provide modest improvements. Given the effectiveness of data augmentation in classification, we should explore the potential solutions for regression.
>
> In the final version of the paper, we will expand on how a method for robust regression method that builds on the causality literature is reinterpreted as a data augmentation strategy. The overall goal is to get better regressors. ADA relies on AR to augment data samples. As in AR, it uses a linear projection to select perturbation directions that are considered more relevant based on the similarity in the batches. Hence, it inherits the generalization properties and prediction robustness (concerning those directions) from AR. Combining the original data with perturbed augmentations in the prescribed directions leads to improved predictions on the mixture of distributions, and they improve generalization.
>
> In Appendix Section A.2, we provide further explanations and figures to illustrate how ADA augments the samples and how each parameter influences the augmented samples. By constructing the Anchor matrix A from a one-hot encoding, ADA “mixes” the samples which fall into the same clusters, and the augmented samples are either moved towards or away from the centroid based on the value of $\gamma$. We control the range of values for $\gamma$ via the hyperparameter $\alpha$.
>
> We made a comparison to the methods proposed in the C-MixUp paper. We also compare ADA with the original MixUp and Mani-MixUp. Additionally, we have added the comparison to the Local MixUp in the attached PDF. Also, the other methods are for classification problems, and their application to regression is not straightforward.
>
> Our proposed method has three hyperparameters: the number of clusters $k$, $\alpha$, and whether to use manifold augmentation. For reference, C-Mixup also has three hyperparameters: bandwidth, $\alpha$ in Beta distribution, and whether to use manifold mixing.
>
> Categorical variables are treated as reals, which makes them meaningless. But small values of gamma might still provide robustness to the solutions. We have not tried to use categorical variables only and wouldn’t expect improvements.
>
> We show how the results vary with $\alpha$ and $k$ for three representative datasets (see the attached PDF). The value of $k$ does not affect the solution significantly, and large $\alpha$ penalized the results.

---

> > ### Comment · Reviewer_dYVk · 2023-08-15
> > **Thank you for the response.**
> >
> > Thanks again for addressing my questions. I will update the score.

---

### Official Review · Reviewer_KQFt · 2023-07-12

**Soundness:** 2 fair
**Presentation:** 2 fair
**Contribution:** 2 fair
**Rating:** 5
**Confidence:** 4

**Summary:**

The paper proposes a new data augmentation method called Anchor Data Augmentation (ADA) that can improve the performance of machine learning classifiers, especially in over-parameterized settings.

Novelty:

The work presents the Anchor Data Augmentation (ADA) which is based on the concept of Anchor Regression. This is a unique approach, but its departure from existing methods is somewhat moderate. The main novelty lies in its application to machine learning classifiers using the concept of linear regression model fitting. However, one could argue that using regression for data augmentation is not entirely groundbreaking.

Significance:

The significance of the paper is somewhat moderate. The idea of using Anchor Regression to improve machine learning classifiers is promising, especially when it comes to over-parameterized settings. However, the paper seems to be more of a proof-of-concept, as it primarily focuses on housing datasets, leaving other domains unexplored. The lack of significant improvement over other existing methods also somewhat diminishes its overall impact.

Soundness:

The methodology presented is fairly sound, with ADA being applied to various types of regression models. However, there are concerns related to the many hyperparameters introduced and the lack of study on their impact, which could be a potential flaw in the work. Additionally, The paper performs a replication-like operation on the data, so isn't there more homogeneous data? The paper does not seem to address potential issues of overfitting in the presence of homogenous data, which is a common concern in machine learning.

Clarity:

The paper is written in a comprehensible manner. It provides a clear explanation of the proposed ADA method and its underlying concept of Anchor Regression. The experiments are also described clearly, though there is room for improvement in detailing the parameter settings and choices made during experimentation. It might benefit from restructuring certain sections to make the presentation even clearer, especially when discussing the results.

Literature:

The paper discusses some existing works, mainly focusing on regression models. However, it would have been beneficial if it drew more comparisons with other data augmentation methods and their results. It feels like there could be a more comprehensive discussion on how ADA fits in the broader landscape of data augmentation techniques, especially in terms of strengths and weaknesses.

Conclusion:

After taking the Area Chair's feedback into consideration, the paper's novelty is evident, but not groundbreaking. Its significance is moderate, given the specific domain application and the moderate improvement over existing methods. Soundness has potential issues due to hyperparameters and the risk of overfitting. The clarity is fairly good but could be improved upon, and there is room for a more in-depth literature review. Based on these updates, my recommendation change to 'Borderline accept' at this time.

**Strengths:**

ADA is based on the concept of Anchor Regression, which involves fitting a linear regression model to a subset of the training data and using the model to predict the target variable for the remaining data.

ADA can be applied to various types of regression models, including neural networks, and can be used to improve predictions in the low data regime.

ADA can be applied to real-world datasets, such as the California and Boston Housing datasets, and can provide improved performance as the number of samples is increased.

**Weaknesses:**

The paper includes many hyperparameters, but does not study the impact of different hyperparameters on the performance of ADA or develop methods for automatically tuning them.

While proposing the ADA method, the paper does not further consider new data augmentation methods that combine ADA with other techniques such as generative adversarial networks or transfer learning.

**Questions:**

How does ADA compare to other data augmentation methods in terms of computational efficiency?

Have you considered applying ADA to other types of classifiers beyond regression models, such as decision trees or support vector machines?

Have you evaluate the robustness of ADA to different types of data distributions and noise levels？

Can it be used for augmentation performance testing on other visual datasets or point cloud datasets?

Why are the hyperparameters set to a uniform distribution in the experiments, and although it's claimed that this is also effective for a gamma distribution, why are no experimental results provided? Is this also effective for other distributions such as chi-square, Rayleigh, sub-Gaussian, and Super-Gaussian ?

**Limitations:**

The paper only explores the application of ADA to housing datasets and does not investigate its use in other domains such as healthcare or finance.

Compared to other methods, the results of this paper's approach did not show significant improvements.

The proposed method might struggle to address the issue of overfitting that tends to occur in scenarios with an abundance of homogenous data.

This is an extremely heuristic paper.

---

> ### Author Rebuttal · Authors · 2023-08-09
>
> Thank you for the constructive comments and discussion about scalability and intuition on hyperparameter selection.
>
> The standard deviations of RMSE and MAPE results are in Tables 1 and 2 in Appendix B.3.
>
> The performance of ADA is indeed dependent on the choice of the anchor matrix A. As with AR, we can choose the anchor matrix based on expert opinion about “plausible” or “importance” shifts (invariances) in the distribution. If not available, we can infer that information from the data. One way to select the anchor matrix is to use the one-hot encoding of cluster membership. This method partitions the data into similar clusters to account for locality and contracts or expands the samples around the cluster centroids. We provide example models trained on the nonlinear (cosine) data when A is defined by varying cluster sizes (Appendix A.2 Figure 4). For larger clusters, more samples are “mixed” to produce one augmented instance, which may result in over-smoothing of the underlying X-Y relation. On the other hand, small clusters lead to less “mixing” and in the extreme case where each cluster has only one member the augmented dataset would be the same as the original. In the general form, $\Pi_A$ provides a “collective” mixing for the samples in a mini-batch to determine a center, and $\gamma$ controls the amount of contraction or expansion around the center.
>
> Therefore, when data has (approximately) linear structure, with sufficiently large clusters the one-hot-based centroid is close to the underlying hyperplane, and the cluster contraction (expansion) would preserve the underlying hyperplane. On the other hand, augmenting data with uniformly distributed perturbation directions would still preserve the underlying hyperplane. We believe this is the case with the California Housing dataset. However, for a nonlinear data structure, augmenting data samples in arbitrary directions with the same perturbation magnitude is bound to fail, and more complicated schemes are necessary.
>
> Data augmentation for regression is hard. There are very few papers, and the gains are typically small. We (as a community) have not been able to find a data augmentation for regression that is as good as for classification. When the gains are small, we should not expect them to be always better than the baseline. But in some cases, we are significantly better, and ADA never hurts the results.
>
> Yes, ADA provides a systematic method to “mix” multiple samples based on collective similarity criteria,  which is measured by the anchor matrix A (which we compute in ADA via a clustering approach). When using k-means clustering to generate A, ADA “mixes” the samples which fall into the same clusters. The cluster centroid is computed via the $\Pi_A$ operator, and augmented samples are moved towards or away from the centroid based on the value of $\gamma$ (see Appendix A.2 “Anchor Matrices and Locality” for details).
>
> We have not found limitations when using our procedure in high dimensional data, and the linear example at the beginning of the experimental section can illustrate what happens as we decrease the number of data points. The error is too high, and no method helps when there is little data. We obtain the same solution with all the algorithms when there is a lot of data. ADA helps in the middle range on the number of data points when there is not too much or too little data. We will never be in the too-much-data regime in high dimensions and should expect to be in the middle range. But we can also be in the too-little-data regime. In that case, it will be hard to find a good regressor unless additional information about the invariances is available.

---

> > ### Comment · Reviewer_KQFt · 2023-08-19
> >
> > Thanks to the author for their work and patience in answering.

---

> ### Comment · Area_Chair_q5ow · 2023-08-18
> **Acknowledging the rebuttal**
>
> Dear reviewer,
>
> Thank you for your time and effort.
>
> The authors have tried to address you comments in their rebuttal.
> What do you think about their response?
>
> Could you please acknowledge the rebuttal as well as the other reviews.
>
> Best,
>
> The AC

---

### Official Review · Reviewer_SE5E · 2023-07-12

**Soundness:** 3 good
**Presentation:** 3 good
**Contribution:** 3 good
**Rating:** 6
**Confidence:** 3

**Summary:**

In this paper, the authors proposed anchor data augmentation, which borrows from the recently proposed Anchor regression method. The anchor data augmentation uses several replicas of the samples, generated according to the anchor matrix. The proposed augmentation is empirically evaluated both for linear and non-linear models.

**Strengths:**

1. The proposed method is simple and easy to implement. The anchor matrix can be generated by k-means clustering without prior knowledge of the data distribution.

2. The illustrative examples of augmenting the data generated from a cosine model show that the proposed method can generate augmented samples that better represent the underlying data distribution.

3. The proposed method is empirically evaluated on both linear and non-linear models. The results on the linear synthetic data with MLP show that the proposed augmentation can significantly improve the generalization performance.

**Weaknesses:**

1. The standard deviation of empirical evaluations is not reported for sections 4.2 and 4.3. It is hard to verify the statistical significance of the results.

2. It seems the choice of $\gamma$ and design of the anchor matrix $A$ largely remain as hyper-parameters and requires tuning. It would be interesting to see some theoretical justification (or intuition) for choosing $\gamma$ and designing $A$.

3. The improvement over vanilla augmentation is somewhat marginal. It seems that for linear synthetic data with the linear model, California Housing dataset, it is not clear whether the proposed method is significantly better than vanilla augmentation.


**Questions:**

When using k-means clustering to generate the anchor matrix, can it be intuitively understood as the augmentation is trying to construct local interpolations to create augmented samples (where the locality is defined by the clusters)? If so, does this method generalize to high-dimensional data well?

**Limitations:**

Yes.

---

> ### Author Rebuttal · Authors · 2023-08-09
>
> Thank you for the constructive comments and discussion about scalability and intuition on hyperparameter selection.
>
> The standard deviations of RMSE and MAPE results are in Tables 1 and 2 in Appendix B.3.
>
> The performance of ADA is indeed dependent on the choice of the anchor matrix A. As with AR, we can choose the anchor matrix based on expert opinion about “plausible” or “importance” shifts (invariances) in the distribution. If not available, we can infer that information from the data. One way to select the anchor matrix is to use the one-hot encoding of cluster membership. This method partitions the data into similar clusters to account for locality and contracts or expands the samples around the cluster centroids. We provide example models trained on the nonlinear (cosine) data when A is defined by varying cluster sizes (Appendix A.2 Figure 4). For larger clusters, more samples are “mixed” to produce one augmented instance, which may result in over-smoothing of the underlying X-Y relation. On the other hand, small clusters lead to less “mixing” and in the extreme case where each cluster has only one member the augmented dataset would be the same as the original. In the general form, $\Pi_A$ provides a “collective” mixing for the samples in a mini-batch to determine a center, and $\gamma$ controls the amount of contraction or expansion around the center.
>
> Therefore, when data has (approximately) linear structure, with sufficiently large clusters the one-hot-based centroid is close to the underlying hyperplane, and the cluster contraction (expansion) would preserve the underlying hyperplane. On the other hand, augmenting data with uniformly distributed perturbation directions would still preserve the underlying hyperplane. We believe this is the case with the California Housing dataset. However, for a nonlinear data structure, augmenting data samples in arbitrary directions with the same perturbation magnitude is bound to fail, and more complicated schemes are necessary.
>
> Data augmentation for regression is hard. There are very few papers, and the gains are typically small. We (as a community) have not been able to find a data augmentation for regression that is as good as for classification. When the gains are small, we should not expect them to be always better than the baseline. But in some cases, we are significantly better, and ADA never hurts the results.
>
> Yes, ADA provides a systematic method to “mix” multiple samples based on collective similarity criteria,  which is measured by the anchor matrix A (which we compute in ADA via a clustering approach). When using k-means clustering to generate A, ADA “mixes” the samples which fall into the same clusters. The cluster centroid is computed via the $\Pi_A$ operator, and augmented samples are moved towards or away from the centroid based on the value of $\gamma$ (see Appendix A.2 “Anchor Matrices and Locality” for details).
>
> We have not found limitations when using our procedure in high dimensional data, and the linear example at the beginning of the experimental section can illustrate what happens as we decrease the number of data points. The error is too high, and no method helps when there is little data. We obtain the same solution with all the algorithms when there is a lot of data. ADA helps in the middle range on the number of data points when there is not too much or too little data. We will never be in the too-much-data regime in high dimensions and should expect to be in the middle range. But we can also be in the too-little-data regime. In that case, it will be hard to find a good regressor unless additional information about the invariances is available.

---

> > ### Comment · Reviewer_SE5E · 2023-08-20
> > **Response**
> >
> > Thank you for answering my questions. I have read the response. I keep my current score unchanged.

---

> ### Comment · Area_Chair_q5ow · 2023-08-18
> **Acknowledging the rebuttal**
>
> Dear reviewer,
>
> Thank you for your time and effort.
>
> The authors have tried to address you comments in their rebuttal.
> What do you think about their response?
>
> Could you please acknowledge the rebuttal as well as the other reviews.
>
> Best,
>
> The AC

---

### Official Review · Reviewer_K137 · 2023-07-21

**Soundness:** 2 fair
**Presentation:** 3 good
**Contribution:** 2 fair
**Rating:** 3
**Confidence:** 4

**Summary:**

This paper introduces a new data augmentation method, designed for regression problems. The method is based on anchor regression, where new samples are generated via their projection to the normal subspace spanned by the anchors. The method is evaluated on several datasets and compared with ERM and C-Mixup, showing on par performance.


**Strengths:**

The authors address the problem of data augmentation for regression which is an emerging topic in ML, and thus the significance is high. The authors motivate their approach using anchor regression, although I doubt the proposed method and AR are at all related. Thus, originality is questionable, however, I might have missed something crucial.


**Weaknesses:**

In general, the paper seems to be technically sound. However, given Alg. 1 and lines 6-7 in it, it is not clear why Eqs. 3 and 4 are even mentioned in the paper, and what is their relation to the method in practice. Essentially, the authors project the data to the normal subspace spanned by $A$. They motivate it via anchor regression (AR), however, it is clear that results totally depend on the choice of $A$, which is somewhat independent of AR to begin with. Also, the authors use indices $i,j$ around equations 7 and 8, but do not specify the indices roles. Thus, one of the main shortcomings of this work is the justification of the approach and its relation to anchor regression is unclear and should be improved.

Another shortcoming is the evaluation section. In particular, it is unclear to me how you select the ADA model among the various ones you trained. In addition, it seems like grid search was more extensive in ADA in comparison to other baselines. Also, why noise (vanilla DA) is added only to output? I would like to see the results of adding noise to input and input&output. There are no standard deviation results. The results of ADA are often inferior to the baselines.


**Questions:**

See above

**Limitations:**

See above

---

> ### Author Rebuttal · Authors · 2023-08-09
>
> Thank you for your direct review and comments about the clarity of the connection between AR and ADA and the derivation of augmentation equations 7 and 8. We will rewrite these sections to improve the fluency and notations. In particular, ADA uses a linear projection as in AR to select perturbation directions considered more relevant based on the similarity of samples in the batch. The augmentation equations (7-8) are row-wise scaled versions of the modified X, Y in AR (5-6), where the scale preserves the (non)linear relation between X and Y after augmentation. The subindices $i$ and $j$ denote the row and column indices of the corresponding matrices as detailed on line 154 of the paper.
>
> We understand from your review that the connection between AR and ADA is murky. The variables in A are the main point of AR, as they are the ones that allow AR to robustify the regression solution, and they make the connection between AR and ADA. The variables in A are typically not available. We decided to use clustering to infer A, inspired by the solution in the first example in the AR paper. It is a design choice. We use locality consistency to create our A for data augmentation (C-MixUp does something similar). Other choices might be possible, but we believe clustering is a solid universal choice. We will improve this description in the paper.
>
> We did the same hyper-parameter search for all the models and did not go the extra mile to make ours better. The hyperparameters used for the benchmarks were the proposed ones in the C-MixUp paper, where they employ cross-validation with grid-search. Furthermore, we report the results of all the models trained with the mentioned hyperparameters for comparison.
>
> We also report the results from the original C-MixUp paper because we could not replicate their results. In other comparison methods, we got better results than C-MixUp reported in [1].  It is important to notice those differences are small and can be due to factors like initialization, random splitting of the data, or the selected optimization procedure. It is not worrisome, but it is notable.
>
> We also added noise to the X variable in vanilla DA, and the results were significantly worse. We decided not to add it to the attached PDF because it did not add anything. We can send it to the Area Chair if you want to see the results.
>
> The standard deviations of RMSE and MAPE results are in Tables 1 and 2 in Appendix B.3.
>
> You mentioned that ADA is sometimes not better than the baseline in your last comment. Data augmentation for regression is hard. There are very few papers, and the gains are typically small. We (as a community) have not been able to find a data augmentation for regression that is as good as for classification. When the gains are small, we should not expect them to be always better than the baseline. But in some cases, we are significantly better, and ADA never hurts the results.
>
> [1] Yao, H., Wang, Y., Zhang, L., Zou, J. Y., & Finn, C. (2022). C-mixup: Improving generalization in regression. Advances in Neural Information Processing Systems, 35, 3361-3376.

---

> > ### Comment · Reviewer_K137 · 2023-08-11
> >
> > Thank you.

---

### Official Review · Reviewer_zYBi · 2023-07-21

**Soundness:** 3 good
**Presentation:** 3 good
**Contribution:** 3 good
**Rating:** 7
**Confidence:** 3

**Summary:**

The authors propose a new approach for automatic data augmentation in nonlinear regression, which requires no specific domain knowledge and the added computation cost is low. The idea is borrowed from Anchor regression and Mixups. The idea is to first run a k-means on the data, use the cluster memberships to construct anchor variables and then perform a non-linear anchor regression. They illustrate the performance of their methods on linear and non-linear regression problems.

**Strengths:**

The authors did a comprehensive literature review, and it borrows the strength of mixup augmentation and anchor regression. Intuitively, by performing a k-means, it allows the method to borrow strength across samples that are "similar", and the anchor regression further improve the robustness. The experiments also show that the method is especially useful when the number of observations is small.

**Weaknesses:**

1. It is not very intuitive why Anchor regression will help generalization from Section 2.2. I understand the first paragraph, but it is not straightforward from the equation in (3). Why do partial pooling and IV help?

2. There is no theoretical analysis of the performance.



**Questions:**

1. Why do the transformation on the variables but not directly modify the loss function as a nonlinear version of (4)?

2. Any rules on selecting the number of clusters or parameter k? Are the results sensitive to different options?

**Limitations:**

The authors have discussed some limitations of their work in the paper.

---

> ### Author Rebuttal · Authors · 2023-08-09
>
> Thank you for your positive review and questions about our paper.
>
> ADA relies on AR to augment data samples, and it inherits the generalization properties of AR. The main objective of AR is prediction robustness concerning some directions of perturbations and interventions. Therefore, it is natural to expect augmentation with ADA to improve the out-of-distributions accuracy for some distribution shifts at the cost of reducing the in-distribution generalization. We understand AR helps with generalization by looking at Equations 5 and 6 instead of 3 because these equations show the transformation of the data before applying OLS. The original data X and Y are projected on the anchor variables. If the Anchor variables are binary, representing a clustering assignment, $\Pi_A*X$ is the cluster centroid. The modified samples pull the original data X (and Y) toward the cluster center if $\gamma>1$, and repel them if $\gamma<1$. AR only uses one value for $\gamma$, and we use a multitude of gamma in ADA to be robust in different ways. In that way, ADA does something similar to C-MixUp, but with more diversity while mixing the samples, as in each mini-batch, the number of samples from the same cluster changes.
>
> The AR paper has a theoretical analysis in which they prove the benefit of using AR to solve regression problems if there is a shift in the distributions of the anchor variables. But we did not carry out an additional analysis of why ADA provides robustness.
>
> We could have modified the loss functions instead of the samples. But we thought it was more natural to change the samples. It is what AR is already doing, i.e., the AR says that solving Eqn. 4 is equivalent to solving OLS with the transformed data (Eqns. 5 and 6).
>
> We used a held-out dataset to set the number of clusters, as we have not developed a good rule for setting them. In the attached Pdf, we added a figure to show how the number of clusters affects the solution, and the results only change slightly.

---

> ### Comment · Area_Chair_q5ow · 2023-08-18
> **Acknowledging the rebuttal**
>
> Dear reviewer,
>
> Thank you for your time and effort.
>
> The authors have tried to address you comments in their rebuttal.
> What do you think about their response?
>
> Could you please acknowledge the rebuttal as well as the other reviews.
>
> Best,
>
> The AC

---

### Official Review · Reviewer_2wsW · 2023-07-23

**Soundness:** 2 fair
**Presentation:** 3 good
**Contribution:** 2 fair
**Rating:** 5
**Confidence:** 5

**Summary:**

This paper design a new mixup data augmentation algorithm for regression problems, which is a challenging field for data augmentations. Specifically, the authors extend Anchor Regression (AR) method as Anchor Data Augmentation (ADA), which utilizes several replicas of the modified samples in AR to generate more training samples. Since AR can provide the data in optimal least square errors once the values of the anchors are known, the proposed ADA generates robust augmented data in the homogeneous groups (achieved by clustering). With comprehensive comparison experiments on linear and non-linear regression problems, the proposed ADA achieves competitive performances with previous mixup approaches, especially C-Mixup (a special-designed mixup method for regression).

**Strengths:**

* **S1**: This paper tackles the data augmentation problem in regression tasks, which is not well-studied in the data augmentation community. Despite the proposed ADA using an existing AR method (refer to W1), this paper still has some valuable contributions and is interesting.

* **S2**: The designed methods are well-motivated and well-written, which is easy to follow. Meanwhile, the experimental settings and hyper-parameters of ADA are provided in the main text or the appendix, which ensures practical usage and reproducibility.

**Weaknesses:**

* **W1**: In comparison to C-Mixup and AR, the novelty of the proposed ADA is relatively weak. The idea of ADA is straightforward, which generates reliable augmented samples in the local scope of the clustered anchors. From my perspective, ADA seems like an improved version of C-Mixup with the existing AR method, which is less novel than these two works.

* **W2**: The conducted experiments mainly focus on 1D data (e.g., linear synthetic data, time series prediction) in tabular format. More data modalities are essential to verify the data augmentation for a generalization purpose. For example, C-Mixup provides angle regression tasks with real-world images.

* **W3**: The compared mixup baselines are not representative enough. For example, C-Mixup compares the learnable mixup methods (e.g., PuzzleMix [1] and AutoMix [2]) in addition to the vanilla mixup and Manifold Mixup. Meanwhile, more lecture review of published mixup augmentation methods is required in Sec. 2.

Considering the pros and cons, I decided to borderline reject this manuscript and look forward to the authors’ feedback during the rebuttal period.

#### Reference
[1] Jang-Hyun Kim, et al. Puzzle Mix: Exploiting Saliency and Local Statistics for Optimal Mixup. ICML, 2020.

[2] Zicheng Liu, et al. AutoMix: Unveiling the Power of Mixup for Stronger Classifiers. ECCV, 2022.

**Questions:**

Refer to weaknesses, and I have no more questions.

**Limitations:**

I have found no more limitations besides my weaknesses and my concerns.

### Post-rebuttal
Thanks for the detailed replies, which addressed my concerns, and I raised the rating to 5. I hope the authors add more experiments and discussions with existing works. Maybe the automatic version of the proposed ADA with fewer hyper-parameters can be the improvement direction.

---

> ### Author Rebuttal · Authors · 2023-08-09
>
> Thank you for your balanced review and your suggestions for improvement.
>
> The literature on Mixup methods is vast and especially interesting for image classification. There are only two valid approaches for regression (reviewed in our paper). C-Mixup is a solid paper, and it brought some of the Mixup gains to regression problems. Anchor regression (AR) is a method to robustify regression methods in a narrow set of cases. It builds on the causality literature and is not a method for data augmentation per se. ADA (our proposal) has several relevant contributions:
> 1 We reinterpret AR as a data augmentation. AR and MixUp come from different sets of assumptions and are almost different research fields in ML altogether. Noticing that AR applies to data augmentation is not straightforward and, in our opinion, a significant contribution.
>
> 2 Our work empirically shows that using $\gamma<1$ helps with generalization too. In MixUp, the augmentation interpolates samples between the original samples. ADA does that for $\gamma > 1$ and pushes them further apart for $\gamma < 1$. It is an added benefit when using the AR framework for data augmentation in regression.
>
> We repeat all the experiments from Sections 5.1 and 5.3 from the C-mixup paper. There are tabular, time series, and image data in that set of experiments. We report the results for the missing datasets in our original submission in the added Pdf file (Table 2). The experiments in Section 5.2 are not reproducible with their publicly available code (comparisons were not possible). Except for the toy cosine data, all our experiments are multi-dimensional.
>
> PuzzleMix and AutoMix are solid contributions to the MixUp literature for image classification. We will add them to our literature review. Both methods are not directly applicable to regression problems. In any case, we have also tried to run the code, it was not straightforward, and we cannot present preliminary results here. In the attached PDF, we have also added Local MixUp for comparison (Tables 1 and 2).

---

> > ### Comment · Reviewer_2wsW · 2023-08-16
> >
> > Thanks for the detailed replies and for conducting additional experiments in PDF. I thought the replies addressed my concerns, and I raised the rating to 5. I believe that improving mixup augmentations for regression tasks is a promising research topic, and I hope the authors add more experiments and discussions with existing works. Maybe the automatic version of the proposed ADA with fewer hyper-parameters can be the improvement direction.

---

### Author Rebuttal · Authors · 2023-08-09

We thank all reviewers for their constructive feedback and suggestions for improving our contribution to NeurIPS, and we hope to have a fruitful discussion in the coming days. We also want to thank the Area Chair for leading this paper review and the discussion. We address each question in our direct responses to the respective review. This global response contains a PDF attachment, in which we present some additional experimental results, which we reference to throughout our responses. First, we show how the ADA results change with different values for $\alpha$ and $k$. Second, we compared the performance of Local Mixup [1] to our baselines and ADA. Last, we added results for two additional datasets to complete the comparison with all the data used by C-MixUp. We are looking forward to the coming discussion with all the reviewers.

[1] Baena, R., Drumetz, L., & Gripon, V. (2022). Preventing manifold intrusion with locality: Local mixup. arXiv preprint arXiv:2201.04368.

---

### Decision · Program_Chairs · 2023-09-21

**Decision:**

Accept (poster)

**Comment:**

I have read the reviews and discussions. I currently tends to agree with the majority of the reviewers that the contribution is valuable since designing good data augmentations for regression is still an open and promising topic.

There were some points raised by reviewer K137 about the impact of the matrix A on the results and on the connection between the paper and AR. From the authors response, I infer that the proposed approach for choosing A by clustering seems to work well in practice and the authors further clarify the connection between AR and ADA. This should be made clearer in the paper.

The experiments appear to be in favor of the method, although the improvements is sometimes marginal compared to other data augmentation methods. Hence, more the conclusions on the method should be more nuanced.

In addition, there were several clarifications that were made during the discussion phase and that must be included in the final version of the paper.